# VisionLLM v2: An End-to-End Generalist Multimodal Large Language Model for Hundreds of Vision-Language Tasks

**Jiannan Wu**[*,2,1]**, Muyan Zhong**[*,3]**, Sen Xing**[*,3]**, Zeqiang Lai**[*,4]**, Zhaoyang Liu**[*,5,1]**, Zhe Chen**[*,6,1]**, Wenhai Wang**[*,7,1]**, Xizhou Zhu**[3,8,1]**, Lewei Lu**[8,1]**, Tong Lu**[6]**, Ping Luo**[2]**, Yu Qiao**[1]**, Jifeng Dai**[†,3,1]

[1]OpenGVLab, Shanghai AI Laboratory    [2]The University of Hong Kong    [3]Tsinghua University
[4]Beijing Institute of Technology    [5]The Hong Kong University of Science and Technology
[6]Nanjing University    [7]The Chinese University of Hong Kong    [8]SenseTime Research

https://github.com/OpenGVLab/VisionLLM

## Abstract

We present VisionLLM v2, an end-to-end generalist multimodal large model (MLLM) that unifies visual perception, understanding, and generation within a single framework. Unlike traditional MLLMs limited to text output, VisionLLM v2 significantly broadens its application scope. It excels not only in conventional visual question answering (VQA) but also in open-ended, cross-domain vision tasks such as object localization, pose estimation, and image generation and editing. To this end, we propose a new information transmission mechanism termed "super link", as a medium to connect MLLM with task-specific decoders. It not only allows flexible transmission of task information and gradient feedback between the MLLM and multiple downstream decoders but also effectively resolves training conflicts in multi-tasking scenarios. In addition, to support the diverse range of tasks, we carefully collected and combed training data from hundreds of public vision and vision-language tasks. In this way, our model can be joint-trained end-to-end on hundreds of vision language tasks and generalize to these tasks using a set of shared parameters through different user prompts, achieving performance comparable to task-specific models. We believe VisionLLM v2 will offer a new perspective on the generalization of MLLMs.

## 1 Introduction

Multimodal large language models (MLLMs) [8, 97, 223, 107, 105, 140, 14, 169, 34, 33] have recently made significant progress, demonstrating outstanding performance across various vision-language tasks, even in scenarios requiring complex understanding and reasoning. However, *a notable limitation is that current MLLM outputs are in text form, which significantly constrains their capacity to represent structured or visual information.* Some researchers [140, 182, 181, 180] have expanded the text-based output formats of MLLMs to better align with downstream tasks. While these efforts have shown promise, they have not fully addressed practical needs such as dense object detection, pose estimation, and image generation.

To overcome this limitation, a line of research [116, 187, 166, 111, 117, 47] enhances the capabilities of MLLMs by transmitting task information to tools via text messages, as illustrated in Figure 1(a). Despite these advances, these text-based methods are restricted by the information that text can convey. They are not end-to-end, and the feedback gradient from the tools cannot be relayed back

---

*Equal contribution. † Corresponding to Jifeng Dai <daijifeng@tsinghua.edu.cn>.

38th Conference on Neural Information Processing Systems (NeurIPS 2024).

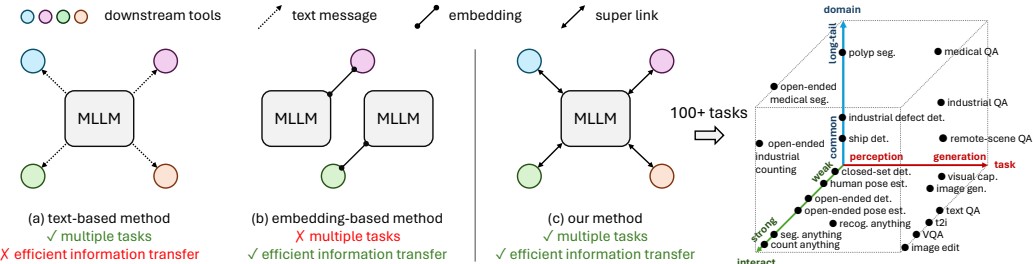

Figure 1: **Illustration of three information transmission methods.** (a) Text-based method shows MLLM connected to various downstream tools via text messages, capable of handling multiple tasks but suffering from inefficient information transfer. (b) The embedding-based method displays a connection using learnable embeddings, which facilitates efficient information transfer but lacks support for multitasking. (c) Our method employs a "super link" technique, where a unified MLLM interfaces with multiple task decoders through super links, supporting over 100 diverse tasks.

to the MLLM. This limitation has spurred another research direction [89, 148, 193, 44, 83, 164] that employs learnable embeddings as intermediaries to connect MLLM with one specific task decoder (see Figure 1(b)). However, the naive embedding connection is difficult to scale to multi-task scenarios. A routing mechanism is needed to ensure the correct selection of tools, and the issue of task conflicts [224] arising from joint multi-task training is also a problem that needs to be considered. Therefore, *developing an end-to-end MLLM generalist for various vision and vision-language tasks beyond text output remains a significant challenge.*

Given these challenges, developing an end-to-end generalist MLLM requires a more effective information transmission method than conventional text messages and naive embeddings. This method should ensure that task information and feedback gradients are accurately and flexibly communicated between the central MLLM and multi-task decoders while preventing task conflicts across various visual domains and input/output formats. In addition, multi-task datasets for generalist MLLMs need to be well-prepared. Despite the abundance of annotations in the community, the diverse and inconsistent formats of these annotations across different tasks make it challenging to develop a unified dataset that effectively supports multi-task learning.

In this work, we introduce VisionLLM v2, an end-to-end generalist MLLM designed for a wide array of vision and vision-language tasks. This model not only performs typical visual question answering but also extends to image generation, image editing, and open-ended object detection/instance segmentation/pose estimation across diverse image domains. To facilitate information transmission between the MLLM and multiple downstream task decoders, we introduce the **super link** technique, which consists of two components: (1) *Routing Token*: special tokens (*e.g.*, [DET], [POSE], and [GEN]) added to the MLLM's vocabulary. Whenever the MLLM predicts a specific routing token, it triggers the selection of the appropriate decoder. (2) *Super-Link Queries* randomly initialized learnable weights bound to the routing tokens. These queries are appended after the routing tokens and processed by the MLLM to extract task-specific information, which is then sent to the target decoder. This method enables flexible task information transmission, allows decoder gradients to backpropagate to the MLLM, and avoids task conflicts by ensuring the queries are bound to routing tokens and not shared across tasks.

Furthermore, we carefully collected and curated training data from hundreds of public vision and vision-language tasks to support various tasks. The data includes high-quality examples of visual question answering, visual perception, recognition, and understanding tasks from various sources such as natural scenes, remote sensing images, medical images, and industrial images. To ensure effective training with these extensive datasets, we also implemented a multi-stage joint training strategy, integrating new abilities and reaching a performance comparable to the expert models while maintaining the MLLM's foundational VQA capabilities.

These designs endow VisionLLM v2 with three distinct characteristics: (1) *Generality*. With one suit of parameters, our model can be generalized to different tasks using different text and visual prompts. To our knowledge, it is the first end-to-end model to support hundreds of vision-language tasks while achieving performance comparable to expert models. (2) *Openness*. By employing open-ended decoders, our model allows users to freely define tasks through multimodal prompts, breaking away from the constraints of closed-set models limited to predefined tasks or categories. Furthermore,

users can flexibly combine various tasks into more complex ones through multi-round dialogue. (3) *Multimodal In-Context Ability*. With multimodal inputs and outputs, our model demonstrates extensive versatility and exhibits superiority over the previous in-context models with single-modal outputs [184, 8]. These features distinguish our model from previous approaches, and establish a leading foundational MLLM for various vision and vision-language applications.

In summary, our main contributions are listed as follows:

(1) We propose VisionLLM v2, the first end-to-end generalist MLLM model to accomplish hundreds of vision and vision-language tasks[1], covering visual perception, understanding, and generation. It not only addresses the limitation of LLMs being confined to text outputs but also supports using textual, visual, and in-context instructions to flexibly combine tasks for real-world applications.

(2) We introduce the super-link technique, which integrates the MLLM with task-specific decoders. This integration facilitates end-to-end optimization across both linguistic and visual tasks. Additionally, we meticulously collect and re-organize data from a broad range of domains and develop an in-context learning dataset. These efforts lay a solid foundation for our progressive joint training process and enable the model to benefit from individual tasks.

(3) We comprehensively evaluate the proposed model on a wide range of vision and vision-language tasks, from visual perception to visual understanding, from weak interaction (*e.g.*, closed-set) to strong interaction (*e.g.*, visual prompt + language prompt), from common-seen domains to long-tailed domains (*e.g.*, medical, remote-sensing, industry), as shown in the rightmost subfigure of Figure 1. In addition, with a generalist model, our method achieves comparable performance with the task-specialized models in various standard benchmarks.

## 2 Related Work

### 2.1 Multimodal Large Language Model

**Conventional MLLMs**. With the advancement of large language models (LLMs) [145, 146, 21, 215, 171, 37, 172, 13, 170, 9, 103, 7, 59, 43, 22], multimodal large language models (MLLMs) have also gained significant momentum recently. Notable commercial models include GPT-4V [2], Gemini series [169, 150], Claude-3 [10], and Qwen-VL-Max [14], known for their outstanding performance. Early open-source MLLMs like InstructBLIP [42], LLaVA [107] and MiniGPT-4 [223] fine-tune on instruction-following datasets. InternVL [34, 33] series models align a large-scale vision encoder with LLMs and perform comparably to commercial models. Efficient MLLMs [100, 228, 38] have also studied. However, these models only can output text, restricting their applications.

**Extension of MLLMs' Text Output**. To extend MLLMs to downstream tasks, models like Kosmo-2 [140], Shikra [27], VisionLLM [182], Ferret [201, 212], and All-Seeing V2 [180] achieve this using specially-designed tokens or encoding coordinates as text tokens. Despite these advancements, using LLMs solely as visual decoders falls short of resolving the fine-grained visual context needed for precise detection and segmentation. The other line of works focus on broadening the modality scope. AnyGPT [210] builds a multimodal text-centric dataset for any-to-any multimodal generation (text, image, speech, music) with sequence modeling. Chameleon [168] uses fully token-based representations for both texts and images, capable of understanding and generating interleaved image-text sequences. CM3leon [5, 205] are autoregressive models for text-to-image and image-to-text tasks. All these works could unify image understanding and generation in one network. Our model can support more vision and vision-language tasks.

**MLLMs w/ Downstream Tools**. Recent works [116, 187, 166, 111, 117, 47, 17, 191, 68, 48] have integrated external tools for vision-centric tasks, transmitting task information to these tools via text messages. However, such text-based communication between LLMs and tools hinders end-to-end optimization. Another category of approaches [89, 148, 218, 83, 164, 163, 53, 54, 136, 44, 50, 69] feeds the output embeddings of LLMs into a special decoder and trains them end-to-end to enhance information communication. However, they only support semantic segmentation or image generation tasks. In this work, we target to develop an end-to-end MLLM generalist for diverse vision and vision-language tasks beyond text output.

---

[1]We consider tasks such that those with differing input and output formats, or those involving data from different domains as distinct tasks.

## 2.2 Vision Generalist Model

**Unified Vision Model.** The unified model approach integrates multiple visual tasks into a single framework, enhancing efficiency and reducing the complexity of deploying separate models for each task. Works such as Pix2Seq-D [29], SEEM [230], and Semantic-SAM [93] focus on unifying the segmentation interface, achieving promising results. Grounding-DINO [112] and VisionLLM [182] explore open-set detection grounded by language, while UniPose [198] excels in pose estimation. Additionally, pioneering works [227, 224, 95, 121, 229, 189] aim to design a unified model capable of solving multiple tasks, including detection, segmentation, captioning, *etc*. Their results demonstrate the feasibility of a single model performing diverse tasks.

**Visual Prompting.** Visual prompting has emerged as a novel paradigm by providing visual marks in the input instruction. It requires the model to pay attention to the specific region on the image when answering the question. Techniques like red circle [157], SoM [196], AutoVP [173], ILM-VP [24], and PIVOT [131] significantly reduce the need for textual prompt engineering, assisting models in focusing on relevant visual content. Similar to in-context learning in LLMs, Painter [183], DINO v2 [91], and SegGPT [184] leverage visual context to improve learning efficiency and adaptability, enabling models to adapt to new tasks with minimal input.

**Diffusion Model as Interface.** Diffusion models are a flexible interface between users and visual tasks, facilitating a more intuitive interaction paradigm. InstructCV [51] and InstructDiffusion [56] exemplify using of natural language instructions to guide visual generation and manipulation. Pix2Seq v2 [30] showcases the potential of diffusion models in generating sequences of visual tokens, bridging the gap between vision and language.

Different from these works, our VisionLLM v2 integrating LLMs extends vision generalist to support a broader range of vision-language tasks and explore various visual prompting paradigms, thereby significantly broadening the scope of application.

# 3 VisionLLM v2

## 3.1 Model Design

The overall architecture of VisionLLM v2 is depicted in Figure 2. It mainly consists of four parts: (1) an image encoder and a region encoder that encode the image-level and region-level information; (2) a large language model (LLM) that models the multimodal inputs and generates satisfactory textual responses; (3) a series of task-specific decoders for performing downstream tasks; (4) a super link that uses routing tokens and super-link queries for efficient and conflict-free information transmission. We detail each component in the following.

**Tokenization.** VisionLLM v2 is flexible for handling multimodal input. (1) *For text prompts*, we employ the text tokenizer to tokenize them into distinct vocabulary indices, which can be further processed by LLM and result in the text features $F_{\text{text}} \in \mathbb{R}^{L \times C}$, where $L$ denotes the length of input text, and $C$ is the channel dimension of LLM.

(2) *For an image input*, we utilize a pre-trained vision foundation model, such as CLIP [144], to extract image features. Recognizing that current vision models operate the images at a low resolution, we adopt the dynamic resolution approach [33] to process the input images. Specifically, the input image is first automatically matched to an optimal aspect ratio from a predefined ratio set. Subsequently, the image is scaled up to a higher resolution based on the selected aspect ratio and divided into $P$ square patches, each whose resolutions are 336×336. These local patches, along with a 336×336 global image $I_{\text{global}}$, are processed by the image encoder to capture both holistic scenes and fine-grained details, resulting in image features $F_{\text{img}} \in \mathbb{R}^{576(P+1) \times C}$.

(3) *For a visual prompt*, we employ binary masks to flexibly represent the visual prompts, such as point, box, scribble, and mask. To extract the region embedding, we first concatenate the binary mask with the input image along the channel dimension and then process it with three convolutional layers to downsample by a factor of 14 (see appendix for more details). We further augment this feature map by adding the feature map of the global image $I_{\text{global}}$. Finally, grid sampling is used to extract features within the masked regions, and these features are averaged to form the features of the visual prompt $F_{\text{vprt}} \in \mathbb{R}^{1 \times C}$.

**Large Language Model.** Following previous works [107, 213, 60], both the images and visual prompts are projected to the feature space of the LLM. The LLM plays a central role in our model

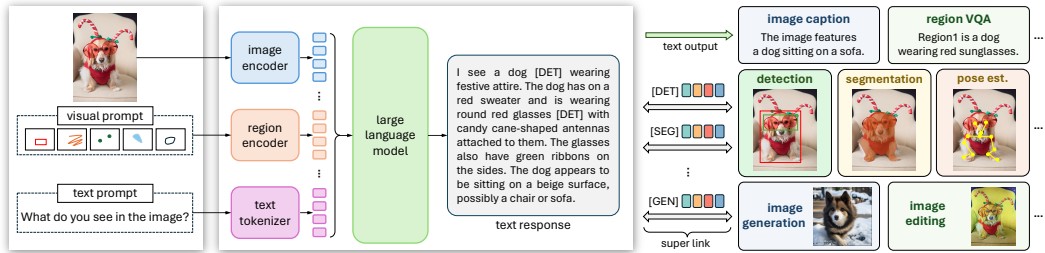

Figure 2: **Overall architecture of the proposed VisionLLM v2.** It receives the image and text/visual prompts as inputs. The central LLM parses the user instructions and generates the textual responses. Besides outputting the plain text, LLM can also output the special routing token such as [DET] when needed. The super-link queries would be automatically appended after the routing token embeddings and further processed by LLM. They play as the bridge for connecting LLM and task-specific decoders. In this way, our generalist model can support hundreds of visual tasks. The detailed architecture about connecting the LLM with task-specific decoders can be found in Figure A13.

and is used to model multimodal inputs, parse user instructions, and generate appropriate responses. In this work, we adopt the commonly used Vicuna-7B [219] as the LLM in our network.

**Task-specific Decoders.** To enhance the capacities of MLLM, we equip our model with several task-specific decoders. Specifically, we use Grounding DINO [112] for object-level localization. We additionally add a mask decoder upon it to obtain the segmentation ability. For pose estimation, we adopt UniPose [198] as the keypoint decoder. Moreover, we incorporate Stable Diffusion [152] and InstructPix2Pix [20] as the image decoders, endowing our model with the capability to generate and edit images. We discard these decoders' text encoders and link them with MLLM via the super link technique, which will be detailed explained in Section 3.2. In this way, the decoders can be trained end-to-end with the entire network, ensuring the effective transmission of task information and increasing the openness of these decoders.

## 3.2 Super Link Technique

For the text-only output tasks, such as image-level and region-level VQA, we directly take the plain text generated by LLM as the final output. For visual perception and visual generation tasks, we propose the super link technique to tackle the challenge of selecting the appropriate decoder, avoiding task conflicts, and facilitating effective information transmission between the LLM and the decoders. The super link comprises two parts:

(1) *Routing Token*. We add the routing tokens, *e.g.*, [DET], [POSE], [SEG], [GEN], [EDIT], as special tokens to the original LLM vocabulary. When the model intends to complete the downstream task using one of the decoders, LLM would include the corresponding routing token in its textual response. To enable the model to discern which tasks to perform and which routing tokens to output, we construct a series of instruction templates for different tasks using ChatGPT [4].

(2) *Super-Link Queries*. For each decoder, we define the super-link queries as a fixed set of embeddings denoted as $Q_{link} \in \mathbb{R}^{N \times C}$, where $N$ is the number of queries. They are randomly initialized and serve as the bridge between LLM and task-specific decoders. Whenever the LLM predicts the routing token, the super-link queries would be automatically appended after the input embeddings of the routing token. We then extract their corresponding last-layer hidden states $H_{link}$ and apply an MLP projection to obtain $\hat{H}_{link}$. Finally, $\hat{H}_{link}$ is sent into the specific decoders as a condition to perform the downstream tasks. In the following, we illustrate how to integrate $\hat{H}_{link}$ into decoders for visual perception and generation, respectively.

**Visual Perception** covers a wide range of visual tasks, such as open-ended/closed-set object detection, instance segmentation, pose estimation, *etc*. VisionLLM v2 supports using both text and visual prompts to define these tasks. We list an example in the following figure. <image> and <region> are the placeholders that will be replaced by image and region embeddings before being fed into the LLM. Here, we take Example 1 of interactive segmentation for clarification. The user prompts the model to segment specific regions within a question. MLLM sequentially lists the region names followed by a routing token [SEG] in the response. Remember that the proposed method would automatically append the super-link queries after the routing token. In that way, we can obtain the

per-region representations by extracting the output hidden states of MLLM from corresponding super-link queries and pooling them into one embedding. These embeddings are fed into a segmentation decoder as the conditional feature, requiring only a single forward to produce segmentation results for all regions. In the following, we show a template example for interactive segmentation.

> **Example1: Text Prompt + Visual Prompt for Interactive Segmentation.**
> **USER**: <image> Could you please segment all the corresponding objects according to the visual prompts as region1 <region>, region2 <region>?
> **ASSISTANT**: Sure, these objects are region1 [SEG], region2 [SEG].

**Visual Generation** is also a wide topic covering a number of different tasks, such as generation, editing, variation, personalization, *etc*. In VisionLLM v2, we focus on two fundamental tasks, *i.e.*, text-to-image generation and instruction-based image editing. We use Stable Diffusion v1.5 (SD) as our tool in the text-to-image generation task. We abandon its text encoder and use the output hidden states of the MLLM as the image generation condition for SD. Image editing task [82] can also be accomplished in the same paradigm by using both image and text prompts as inputs. In the following, we list a template example for text-to-image generation.

> **Example 2: Text Prompt for Text-to-Image Generation.**
> **ASSISTANT**: Of course, here it is [GEN].

*Discussion.* Some previous works have used the special token or learnable queries independently. InstructBLIP [42], ep-ALM [158], and MAPL [125] use learnable queries (i.e., soft prompts) to connect the modality encoders and LLM. FROMAGe [84] uses a special token for image-text retrieval so as to handle multimodal outputs, where the images are not generated from the network end-to-end. However, these works still remain constrained to text-based outputs. The proposed super link is the seamless integration of the two techniques. Despite the simplicity of our method, it is able to extend MLLMs to handle hundreds of tasks by largely extending the output formats, *e.g.*, box, mask, keypoint and image. Meanwhile, it can address several challenges when scaling up various tasks: (i) precise decoder invocation, (ii) mitigating task conflicts and (iii) efficient message transmission in an end-to-end manner.

### 3.3 Training Strategy

Current MLLMs [89, 218, 44] face reduced conversational abilities when augmented with additional capacities. To create a generalist model capable of handling hundreds of tasks without compromising vision understanding, we propose a three-stage training strategy, where the first stage focuses on building an MLLM with strong image-level and region-level vision understanding. In the subsequent stages, we add task-specific decoders and continue training to equip the model with advanced capabilities.

**Stage-1: Mutimodal Training.** In the first stage, we follow the training settings of LLaVA [107, 105], comprising pre-training and instruction tuning phases. The pre-training phase aims to establish the image-level and region-level vision-language alignment, where only the region encoder and the projections for image embedding and region embedding are trained for efficiency. The instruction tuning phase unfreezes the LLM and trains the model on a wide range of high-quality instruction data. After the training in this stage, we can obtain a strong MLLM with excellent conversation ability, which we term as **VisionLLM v2-Chat**.

**Stage-2: Multi-capacity Fine-tuning.** At this stage, we integrate task-specific decoders into the model and perform multi-task joint training. In addition to the instruction data utilized in stage-1, we incorporate extensive visual datasets such as COCO [104], ADE20K [222] for their specific tasks. We construct a series of instruction templates for these visual datasets to perform instruction tuning, ensuring that the LLM can accurately invoke the downstream decoders. During this stage, the region encoder and all decoders undergo training, and we only finetune the input and output embeddings of the LLM to maximally preserve its original conversational ability.

**Stage-3: Decoder-only Fine-tuning.** Since the decoders cannot converge within a single epoch, we further train the decoders for 12 epochs using visual datasets while freezing all other components. It is noted that the super-link queries continue to be trained during this stage. After finishing the three-stage training, our model has diverse capacities for visual tasks while maintaining effectiveness in global vision understanding, named **VisionLLM v2**.

| model | visual encoder | LLM | academic-oriented datasets | | | | | instruction-following datasets | | | |
|---|---|---|---|---|---|---|---|---|---|---|---|
| | | | VQA$^{v2}$ | GQA | VizWiz | SQA$^I$ | VQA$^T$ | POPE | MME | MMB-EN/CN | SEED |
| InstructBLIP-7B [42] | EVA-g | Vicuna-7B | - | 49.2 | 34.5 | 60.5 | 50.1 | - | - | 36.0 / 23.7 | 53.4 / 58.8 / 38.1 |
| InstructBLIP-13B [42] | EVA-g | Vicuna-13B | - | 49.5 | 33.4 | 63.1 | 50.7 | 78.9 | 1212.8 | - / - | - / - / - |
| Shikra [27] | CLIP-L | Vicuna-13B | 77.4* | - | - | - | - | - | - | 58.8 / - | - / - / - |
| IDEFICS-80B [72] | CLIP-H | LLaMA-65B | 60.0 | 45.2 | 36.0 | - | 30.9 | - | - | 54.5 / 38.1 | - / 53.2 / - |
| Qwen-VL-Chat [14] | CLIP-G | Qwen-7B | 78.2* | 57.5* | 38.9 | 68.2 | 61.5 | - | 1487.5 | 60.6 / 56.7 | 58.2 / 65.4 / 37.8 |
| InternVL-7B [34] | ViT-6B | Vicuna-7B | 79.3* | 62.9* | 52.5 | 66.2 | 57.0 | 86.4 | 1525.1 | 64.6 / 57.6 | 60.2 / - / - |
| InternVL-13B [34] | ViT-6B | Vicuna-13B | 80.2* | 63.9* | 54.6 | 70.1 | 58.7 | 87.1 | 1546.9 | 66.5 / 61.9 | 62.4 / - / - |
| LLaVA-1.5-7B [105] | CLIP-L | Vicuna-7B | 78.5* | 62.0* | 50.0 | 66.8 | 58.2 | 85.9 | 1510.7 | 64.3 / 58.3 | 58.6 / 66.1 / 37.3 |
| LLaVA-NeXT-7B [106] | CLIP-L | Vicuna-7B | 81.8* | 64.2* | 57.6 | 70.1 | 64.9 | 86.5 | 1519.0 | 67.4 / 60.6 | - / 70.2 / - |
| LLaVA-NeXT-13B [106] | CLIP-L | Vicuna-13B | 82.8* | 65.4* | 60.5 | 73.6 | 67.1 | 86.2 | 1575.0 | 70.0 / 64.4 | - / 71.9 / - |
| VisionLLM v2-Chat | CLIP-L | Vicuna-7B | 81.4* | 65.1* | 54.6 | 94.4* | 66.3 | 87.5 | 1512.5 | 77.1 / 67.6 | 65.4 / 71.7 / 41.6 |
| VisionLLM v2 | CLIP-L | Vicuna-7B | 80.8* | 65.1* | 51.8 | 94.2* | 64.7 | 88.8 | 1495.6 | 76.3 / 66.8 | 65.6 / 71.7 / 42.2 |

Table 1: **Comparison with SoTA models on multimodal dialogue benchmarks.** The academic-oriented datasets include: VQAv2 test-dev [57], GQA test-balanced [71], VizWiz test-dev [62], ScienceQA test [154] and TextVQA val [160]. The instruction-following datasets include: POPE [101], MME [49], MMBench-EN/CN [114], SEED-Bench (all/image/video) [55]. *The training annotations of the dataset are observed during training.

## 4 Experiments

### 4.1 Implementation Details

**Dataset Details.** To support the joint training of our model, we meticulously collect and re-organize the datasets across a wide range of tasks from publicly available sources. For the first stage training, we utilize a substantial amount of high-quality instruction data for both image-level and region-level visual question answering, including ShareGPT4V [28], All-Seeing [181], VQAv2 [57], *etc*. In the last two stages, we further incorporate extensive visual datasets, *e.g.*, COCO [104], RefCOCO/+/g [204, 126], LAION-Aesthetics [3], to enhance our model with numerous capacities. These datasets encompass multiple tasks such as object detection, pose estimation, image generation, and span various domains such as natural scenes, remote sensing images, medical images, *etc*. To facilitate the training of diverse datasets in our MLLM framework, we construct a series of instruction templates for different tasks, which are completely listed in the Appendix. Additionally, we also collect a multimodal dataset termed **MMIC** focusing on visual prompting and in-context learning. The data in our MMIC comes from various sources, including fine-grained visual recognition, object detection, instance segmentation, and pose detection. We elaborate on all datasets used in this work as well as the dataset construction of MMIC in the Appendix.

**Model Details.** We adopt the CLIP-L/14 [144] as the image encoder and Vicuna-7B-v1.5 [219] as the language model. Grounding-DINO [112] and UniPose [198] are selected as object decoder and keypoint decoder, respectively. And for these two decoders, we experiment with Swin-T [115] backbone. Additionally, image decoders are kept as Stable Diffusion v1.5 [152] for image generation and InstructPix2Pix [20] for image editing. All these components load the pre-trained weights while the region encoder is randomly initialized. For visual perception and visual generation tasks, the number $N$ of super-link queries is set to 4 and 64, respectively. During training, we adjust the dataloader so that each GPU processes samples from only one dataset. More training details are provided in the Appendix.

In the following subsections, we present the experimental results to cover as many tasks, interactive modes, and domains. It is noted that all the results of our method are reported using **a single generalist model** with the same parameters. More results can be found in the Appendix.

### 4.2 Mutimodal Benchmarks

**Multimodal Dialogue.** We first evaluate our models on academic-oriented VQA datasets and recent instruction-following datasets for MLLMs, as presented in Table 1. The results clearly demonstrate that our models outperform previous methods under the same parameter scale, particularly on the instruction-following datasets. For instance, VisionLLM v2-Chat surpasses LLaVA-NeXT-7B [106] by +9.7 and +7.0 points on MMBench-EN/CN [114], respectively. Additionally, we find that VisionLLM v2 achieves comparable performance to VisionLLM v2-Chat on these multimodal benchmarks and even performs better on some benchmarks, such as POPE [101], a popular benchmark for evaluating object hallucination. This phenomenon indicates that our framework effectively mitigates the issue of multi-task conflict and maintains proficiency in conversational ability.

| method | COCO | | LVIS | | PACO | |
|---|---|---|---|---|---|---|
| | mAP | Acc (%) | SS | S-IoU | SS | S-IoU |
| CLIP [144] | 58.9 | - | - | - | - | - |
| RegionCLIP [221] | 58.3 | - | - | - | - | - |
| LLaVA [107] | - | 40.0 | 49.9 | 19.8 | 42.2 | 14.6 |
| Shikra [27] | - | 53.9 | 49.7 | 19.8 | 43.6 | 11.4 |
| GPT4RoI [213] | - | 64.0 | 51.3 | 12.0 | 48.0 | 12.1 |
| ASM [181] | 69.3 | - | - | - | - | - |
| RegionGPT [60] | 70.0 | 80.6 | - | - | - | - |
| Osprey [207] | - | - | 65.2 | 38.2 | 73.1 | 52.7 |
| VisionLLM v2-Chat | 81.8 | 90.5 | 67.3 | 42.7 | 63.8 | 36.3 |
| VisionLLM v2 | 81.9 | 90.4 | 73.0 | 51.3 | 70.9 | 47.6 |

(a) Region Recognition

| method | Val Acc (%) | | |
|---|---|---|---|
| | Q→A | QA→R | Q→AR |
| ViLBERT [120] | 72.4 | 74.5 | 54.0 |
| Unicoder-VL [94] | 72.6 | 74.5 | 54.5 |
| VLBERT [162] | 75.5 | 77.9 | 58.9 |
| ERNIE-ViL-L [202] | 78.5 | 83.4 | 65.8 |
| VILLA [52] | 78.5 | 82.6 | 65.2 |
| GPT4RoI-7B* [213] | 87.4 | 89.6 | 78.6 |
| ASMv2 [180] | 87.8 | 88.8 | 78.4 |
| ASMv2* [180] | 88.4 | 89.9 | 79.4 |
| VisionLLM v2-Chat | 90.0 | 91.9 | 82.9 |
| VisionLLM v2 | 89.8 | 91.7 | 82.5 |

(b) Visual Commonsense Reasoning

Table 2: **Comparison of region recognition and visual commonsense reasoning performance.** (a) SS and S-IoU represent semantic similarity and semantic IoU, which originated from [207]. (b) Q, A, and R denote question, answer, and rationale, respectively. X→Y means that the model needs to select option Y conditioned on X. *The model is finetuned on the dataset.

| method | type | backbone | detection (COCO) | | | instance seg. (COCO) | | | detection (CrowdHuman) | | |
|---|---|---|---|---|---|---|---|---|---|---|---|
| | | | AP | AP$_{50}$ | AP$_{75}$ | AP | AP$_{50}$ | AP$_{75}$ | AP$_{50}$ | mMR↓ | Recall |
| Deformable-DETR [226] | Specialist | ResNet50 | 46.2 | 65.2 | 50.0 | - | - | - | 89.1 | 50.0 | 95.3 |
| DDQ [214] | | ResNet50 | 52.0 | 69.5 | 57.2 | - | - | - | 93.8 | 39.7 | 98.7 |
| ViTDet-B [99] | | ViT-B | 56.0 | - | - | 48.0 | - | - | - | - | - |
| Grounding DINO [112] | | Swin-T | 57.2 | - | - | - | - | - | - | - | - |
| Mask2Former [35] | | ResNet50 | - | - | - | 43.7 | - | - | - | - | - |
| Mask DINO [92] | | ResNet50 | 51.7 | - | - | 46.3 | - | - | - | - | - |
| UniHCP* [39] | Generalist | ViT-B | - | - | - | - | - | - | 92.5 | - | - |
| Hulk [185] | | ViT-L | - | - | - | - | - | - | 92.2 | - | - |
| Hulk* [185] | | ViT-L | - | - | - | - | - | - | 93.0 | - | - |
| Pix2Seq v2 [30] | | ViT-B | 46.5 | - | - | 38.2 | - | - | - | - | - |
| VisionLLM [182] | | ResNet50 | 44.8 | 64.1 | 48.5 | 25.2 | 50.6 | 22.4 | - | - | - |
| Uni-Perceiver-v2 [95] | | Swin-B | 58.6 | - | - | 50.6 | - | - | - | - | - |
| UNINEXT [195] | | ResNet50 | 51.3 | 68.4 | 56.2 | 44.9 | 67.0 | 48.9 | - | - | - |
| GLEE-Lite [190] | | ResNet50 | 55.0 | - | - | 48.4 | - | - | - | - | - |
| GLEE-Plus [190] | | Swin-L | 60.4 | - | - | 53.0 | - | - | - | - | - |
| VisionLLM v2 | | Swin-T | 56.7 | 74.5 | 62.2 | 47.8 | 71.8 | 52.0 | 93.1 | 44.7 | 98.5 |

Table 3: **Comparison of object detection and instance segmentation performance.** Instance seg. means instance segmentation. *The model is finetuned on the dataset.

| method | type | backbone | AP↑ | | | | | PCK@0.2↑ | | | |
|---|---|---|---|---|---|---|---|---|---|---|---|
| | | | COCO | CrowdPose | AP-10K | Human-Art | Macaque | 300W | AnimalKingdom | Fly | Locust |
| ViTPose++ [194] | Specialist | ViT-S | 75.8 | - | 71.4* | 23.4 | 15.5* | 95.2* | - | - | - |
| ED-Pose [197] | | Swin-T | 73.3 | - | 45.5 | 71.3 | 51.0 | - | - | - | - |
| UniPose-T [198] | Generalist | Swin-T | 74.4 | - | 74.0 | 72.5 | 78.0 | 98.1 | 95.3 | 99.6 | 99.7 |
| UniPose-V [198] | | Swin-T | 74.3 | - | 73.6 | 72.1 | 77.3 | 99.4 | 94.3 | 99.8 | 99.6 |
| VisionLLM v2 | | Swin-T | 74.0 | 79.4 | 76.8 | 72.9 | 81.9 | 91.1 | 94.4 | 99.4 | 97.8 |

Table 4: **Comparison of pose estimation performance.** * indicates that the results rely on ground-truth bounding boxes for top-down methods.

**Region Recognition.** The region recognition task needs the model to identify the object category given the ground-truth bounding box. We compare our method with both feature-based and text-output approaches in Table 2a. Feature-based methods, such as RegionCLIP [221] and ASM [181], compute similarity scores between region visual features and candidate category text features. In contrast, text-output methods [25, 60, 207] directly predict the category name using a single word or phrase, embracing the advantage of openness. As shown in the table, our models demonstrate the significant superior performance on COCO [104], long-tail LVIS [61] and part-level PACO [147].

**Visual Commonsense Reasoning.** Visual commonsense reasoning (VCR) requires the model to possess strong region-level question-answering and reasoning abilities, as it needs to select not only the correct answer but also the correct rationale behind it. We present the comparison results on the VCR dataset [209] in Table 2b. Without task-specific fine-tuning, VisionLLM v2-Chat achieves an accuracy of 82.9% in the crucial Q→AR task, which precedes the previous best model, ASMv2 [180], by +3.5 points. VisionLLM v2 also outperforms the previous methods for all the metrics, highlighting the promising common sense reasoning capability of our model.

### 4.3 Visual Perception Tasks

**Object Detection and Instance Segmentation.** In Table 3, we compare the results of VisionLLM v2 with state-of-the-art methods on two fundamental vision tasks, *i.e.*, object detection, and instance segmentation. As can be seen, using the lightweight backbone Swin-T, our generalist model achieves

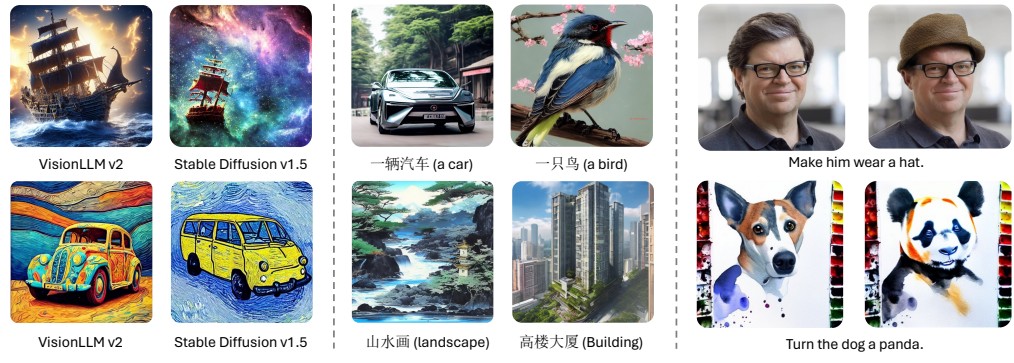

| (a) Text-to-Image Generation | (b) Zero-shot Bilingual Image Generation | (c) Instruction-based Image Editing |

Figure 3: **Qualitative results of image generation and image editing.** The prompts for text-to-image generation are "Pirate ship trapped in a cosmic maelstrom nebula" and "A car in the style of van Gogh."

| method | query/token number | inst seg. $AP_b$ | inst seg. $AP_m$ | ground. P@.5 | pose AP | interact seg. mIoU | interact seg. cIoU |
|---|---|---|---|---|---|---|---|
| | 1 | 50.4 | 39.6 | 85.8 | 43.0 | 43.2 | 60.0 |
| super-link queries | 4 | 52.0 | 41.0 | 85.7 | 71.0 | 44.8 | 60.4 |
| | 8 | 52.1 | 40.7 | 86.4 | 71.6 | 45.9 | 61.9 |

Table 5: **Ablation on the super-link queries number.** We evaluate the results on the four crucial visual perception tasks: instance segmentation (COCO), visual grounding (RefCOCO), pose estimation (COCO), and interactive segmentation (COCO using scribble). Our default setting is marked in gray.

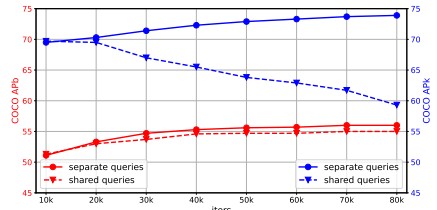

Figure 4: **Shared vs. unshared super-link queries for different decoders.** We report the box/keypoint AP on COCO.

the performance of 56.7 $AP_b$ and 47.8 $AP_m$ on COCO. The results significantly outperform the previous methods using ResNet50 [64] backbone and are comparable with the specialist model Grounding-DINO-T [112]. Moreover, we also validate our model on the crowded pedestrian detection dataset, *i.e.*, CrowdHuman. VisionLLM v2 surpasses the previous best generalist model Hulk [185] by 0.9 points on $AP_{50}$.

**Pose Estimation.** We present the results on the multiple pose estimation dataset in Table 4. While most previous methods [197, 194, 108] only focus on the person scenes, our VisionLLM v2 is effective in performing the keypoint detection for multiple objects. As shown in the table, our model achieves competitive performance with UniPose-T [198] using the same Swin-T backbone. Especially, our model demonstrates the superior performance on AP-10K [203] and Macaque [88] datasets and sets a new state-of-the-art result on CrowdPose [96]. These results prove the effectiveness of our model for pose estimation.

## 4.4 Visual Generation Tasks

We evaluate the generation capabilities of our model on two tasks, *i.e.*, text-to-image generation and instruction-based image editing. In Figure 3, we demonstrate that even if our model uses Stable Diffusion v1.5 as an image decoder, it achieves better visual quality than SD v1.5 with better conditional embedding produced by LLM. Moreover, the use of LLM for conditional encoding of user instructions makes it possible to benefit from the merits of LLM. For example, our model trained on English data is able to perform zero-shot bilingual image generation. Besides, we show the qualitative results of applying our model for instruction-based image editing, which also achieves appealing performance in a unified approach.

## 4.5 Ablation Study

In the ablation studies, we follow the training setup of stage-2 unless otherwise specified. We only train our model on the crucial visual perception tasks, *i.e.*, instance segmentation, visual grounding, pose estimation, and interactive segmentation, for rapid validation.

**Super-Link Queries Number.** We ablate the number $N$ of super-link queries in Table 5. We observe that the performance of these tasks consistently improves with an increasing number of queries. This is reasonable as more queries can lead to richer and stronger representations.

| Train / Test | Image VQA | Inst. Seg | Image Gen. |
|---|---|---|---|
| Image VQA | -0.01 | -0.11 | -0.04 |
| Inst Seg. | +0.04 | -0.12 | + 0.19 |
| Image Gen. | +0.03 | +0.02 | -0.04 |

Table 6: **Ablation on the multi-task influence.** The numbers denote the loss change when the model is fine-tuned on a single task.

| | TextVQA | MME | MMB EN/CN | COCO | COCO-Pose |
|---|---|---|---|---|---|
| One-stage | 53.2 | 1284.4 | 61.9 / 51.4 | 54.9 / 44.6 | 74.1 |
| Three-stage | 66.2 | 1507.1 | 77.8 / 68.5 | 56.3 / 47.6 | 74.2 |

Table 7: **Ablation on the one-stage and three-stage training.** We evaluate the models on image VQA, instance segmentation and pose estimation

**Shared *vs.* Unshared Super-Link Queries for Different Decoders.** To determine if one set of super-link queries is sufficient for all decoders, we conducted an ablation study by either using shared queries for all decoders or defining separate queries for each decoder. In this ablation, we only train the decoders and super-link queries while freezing all other components as the training setting of stage-3. In Figure 4, we plot the performance of box AP (using the object decoder) and keypoint AP (using the keypoint decoder) on COCO. We observe that the keypoint AP would decrease over training when using shared queries, which may be attributed to the fact that most data are used for object decoder. Besides, the box AP with shared queries is also inferior to decoupled ones. Therefore, we define separate super-link queries for each decoder in our model.

**Multi-Task Influence.** As indicated by previous works [225, 206], different tasks with shared parameters may cause conflict with each other. This is mainly due to inconsistent optimization in multi-task learning. To investigate the mutual influence of multi-task joint training in our framework, we start from the same checkpoint and train the model on a single task (image VQA, instance segmentation, or image generation) for 1000 iterations. We record the loss change for all three tasks in Table 6. In the table, a decrease in the loss value indicates beneficial training for the task, while an increase is detrimental. We can observe that training on image VQA is advantageous for all three tasks, which is reasonable as the conversation ability of MLLM is enhanced. Whereas training exclusively on instance segmentation or image generation leads to conflicts with other tasks. This aligns with the findings in Uniperceiver-MoE [225].

**One-Stage *vs.* Three-Stage Training.** Some previous generalist models [176, 15] train the model in one stage. Our model encompasses much more tasks and thus introduces a training conflict: the MLLM requires only 1 epoch of training on chat data to prevent overfitting, whereas the decoders need longer training epochs (e.g., Grounding-DINO need 12 epochs of training on visual data) to achieve convergence. One possible solution for one-stage training is to give a higher sample ratio for the visual data. In the following, we conduct the ablation to study the effect of one-stage v.s. three-stage training. We use image-level chat data, COCO, and COCO-Pose for image understanding, instance segmentation, and pose estimation, respectively. For one-stage training, we repeat the COCO and COCO-Pose datasets 12 times. As can be seen from Table 7, the conversation ability of the model is significantly decreased due to extreme data imbalance. And the performance of instance segmentation and pose estimation is also slightly reduced. These results prove the effectiveness of our three-stage training.

## 5 Conclusion & Limitation

In this paper, we presented VisionLLM v2, a comprehensive MLLM that unifies visual perception, understanding, and generation within a single framework. The proposed super link mechanism facilitates flexible information transmission between the MLLM and task-specific decoders, addressing training conflicts and enhancing gradient feedback. Experiments show that VisionLLM v2 achieves performance comparable to specialized models while maintaining broad applicability.

Regarding limitations, our model's training encompasses three stages, which are relatively complex. Moreover, the integration of downstream tools has only been preliminarily validated. Future work will further explore solutions to these issues, aiming to enhance the model's performance and efficiency.

**Broader Impact.** We envision that this work will further promote the fusion of visual and language tasks. In addition, since our work is built on open-source pre-trained vision foundation models and large language models, requiring low training resources, thus reducing the carbon footprint. We do not foresee obvious undesirable ethical/social impacts at this moment.

# Acknowledgement

This paper is supported by the National Key R&D Program of China (No.2022ZD0161000), the General Research Fund of Hong Kong (No.17200622, 17209324), and the National Natural Science Foundation of China (No. 62376134, 62372223). Tong Lu and Zhe Chen are supported by the China Mobile Zijin Innovation Insititute (No. NR2310J7M). Zhe Chen is also supported by the Youth PhD Student Research Project under the National Natural Science Foundation (No. 623B2050).

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

# The Appendix of VisionLLM v2

## A  More Results

### A.1  More Experimental Results

**Region Captioning.** To access the region understanding capabilities of VisionLLM v2, we evaluate our models on two prominent benchmarks: RefCOCOg [126] and Visual Genome [86]. The results are presented in Table A1b. Notably, VisionLLM v2-Chat significantly outperforms the state-of-the-art methods, with the improvements of +8.6 and +8.4 points in CIDEr scores on RefCOCOg [126] and Visual Genome (validation subset) [86, 148], respectively. The generalist VisionLLM v2 also shows promising performance on RefCOCOg. These results demonstrate the strong fine-grained region captioning capabilities of our model.

**Visual Grounding.** Visual grounding is a crucial vision task that associates the language description with the specific object within an image. Using the box or mask as the output format, visual grounding can be further categorized into referring expression comprehension (REC) and referring expression segmentation (RES) tasks. We comprehensively list the comparison results for the two tasks in Table A2 and Table A3, respectively. From Table A2, it is found that VisionLLM v2 achieves the competitive performance on RefCOCO [204] among MLLMs. We also showcase that VisionLLM v2 exhibits remarkable pixel-level segmentation capacities in Table A3. Without further fine-tuning, our model demonstrates the good gIoU result of 51.0 on the challenging ReasonSeg dataset [89].

**Semantic Segmentation.** In addition to the instance-level segmentation, our model also has the capacity to address the task of semantic segmentation. We present the results on ADE20K [222] in Table A4. The previous works mainly follow the standard training setting for 160k iterations on 8 GPUs with a total batch size of 16. As ADE20K only constitutes a small proportion of our joint-training dataset, our generalist model has a slightly inferior performance on this dataset. By fine-tuning this dataset with fewer training iterations, *i.e.*, 45k, VisionLLM v2 can achieve a mIoU of 52.3 points, surpassing the previous methods under the same backbone.

**Interactive Segmentation.** Interactive segmentation [82] is an emerging task that uses visual prompts as conditions for instance segmentation. We compare our method with state-of-the-art approaches on the COCO-interactive dataset [218] in Table A5. This dataset, proposed by [218], utilizes points, scribbles, boxes, and masks as visual prompts and is annotated on the COCO dataset [104]. As shown in the table, our generalist model VisionLLM v2 demonstrates performance advantages over SEEM-B [230] across all metrics but falls behind the recently proposed MLLM method PSALM [218]. We hypothesize that this is due to our region encoder being frozen during stage-3 of training, which constrains the model's performance. Therefore, we further fine-tune our model on this task by unfreezing the region encoder. It is observed that the performance of our model is significantly improved, as illustrated in the last row of Table A5.

| method | Flickr30K | NoCaps |
|---|---|---|
| Flamingo-80B [8] | 67.2 | - |
| Kosmos-2 [140] | 66.7 | - |
| BLIP-2 [97] | 71.6 | 103.9 |
| InstructBLIP [42] | 82.8 | 121.9 |
| Shikra-13B [27] | 73.9 | - |
| ASM [181] | 87.7 | 117.2 |
| InternVL-G [34] | 79.2 | 113.7 |
| Qwen-VL [14] | 85.8 | 121.4 |
| Qwen-VL-Chat [14] | 81.0 | 120.2 |
| VisionLLM v2-Chat | 88.7 | 118.1 |
| VisionLLM v2 | 90.0 | 116.2 |

| method | RefCOCOg | | VG (full set) | | VG (subset) | |
|---|---|---|---|---|---|---|
| | METEOR | CIDEr | METEOR | CIDEr | METEOR | CIDEr |
| GRiT [188] | 15.2 | 71.6 | 17.1 | 142.0 | - | - |
| Kosmos-2 [140] | 14.1 | 62.3 | - | - | - | - |
| GPT4RoI [213] | - | - | 17.4 | 145.2 | - | - |
| ASM [181] | 20.8 | 103.0 | 18.0 | 145.1 | - | - |
| RegionGPT [60] | 16.9 | 109.9 | 17.0 | 145.6 | - | - |
| PixelLLM [193] | 14.3 | 82.3 | 19.9 | 148.9 | - | - |
| GLaMM [148] | 16.1 | 107.3 | - | - | 19.0 | 163.9 |
| Groma [123] | 16.8 | 107.3 | - | - | 19.0 | 158.4 |
| VisionLLM v2-Chat | 21.2 | 118.5 | 17.8 | 149.2 | 20.0 | 172.3 |
| VisionLLM v2 | 21.1 | 116.6 | 17.5 | 146.7 | 19.8 | 170.1 |

(a) Zero-shot image captioning.  (b) Region captioning.

Table A1: **Comparison of zero-shot image captioning and region captioning performance.** Zero-shot image captioning is evaluated on Flickr30K test set [142] and NoCaps validation set [6], using CIDEr as evaluation metric. For region captioning on Visual Genome [86], full set refers to the use of all validation samples for evaluation, while subset denotes the 5000 samples specified by [148].

| method | type | RefCOCO | | | RefCOCO+ | | | RefCOCOg | |
|---|---|---|---|---|---|---|---|---|---|
| | | val | testA | testB | val | testA | testB | val | test |
| UNITER [32] | VGM | 81.4 | 87.0 | 74.2 | 75.9 | 81.5 | 66.7 | 74.0 | 68.7 |
| VILLA [52] | | 82.4 | 87.5 | 74.8 | 76.2 | 81.5 | 66.8 | 76.2 | 76.7 |
| MDETR [76] | | 86.8 | 89.6 | 81.4 | 79.5 | 84.1 | 70.6 | 81.6 | 80.9 |
| Grounding DINO T* [112] | | 89.2 | 91.9 | 86.0 | 81.1 | 87.4 | 74.7 | 85.2 | 84.9 |
| Grounding DINO L* [112] | | 90.6 | 93.2 | 88.2 | 82.8 | 89.0 | 75.9 | 86.1 | 87.0 |
| Shikra-7B [27] | MLLM | 87.0 | 90.6 | 80.2 | 81.6 | 87.4 | 72.1 | 82.3 | 82.2 |
| Shikra 13B [27] | | 87.8 | 91.1 | 81.8 | 82.9 | 87.8 | 74.4 | 82.6 | 83.2 |
| MiniGPT-v2-7B [26] | | 88.1 | 91.3 | 84.3 | 79.6 | 85.5 | 73.3 | 84.2 | 84.3 |
| Qwen-VL-7B [14] | | 88.6 | 92.3 | 84.5 | 82.8 | 88.6 | 76.8 | 86.0 | 86.3 |
| VistaLLM [143] | | 88.1 | 91.5 | 83.0 | 82.9 | 89.8 | 74.8 | 83.6 | 84.4 |
| Ferret-7B [201] | | 87.5 | 91.4 | 82.5 | 80.8 | 87.4 | 73.1 | 83.9 | 84.8 |
| VisionLLM v2 | | 87.9 | 91.2 | 84.3 | 77.6 | 83.8 | 70.2 | 82.9 | 84.1 |

Table A2: **Comparison of referring expression comprehension performance.** The results are reported based on P@0.5. VGM and MLLM represent vision generalist model and multimodal large language model, respectively. *The model is finetuned on the dataset.

| method | type | RefCOCO | | | RefCOCO+ | | | RefCOCOg | | ReasonSeg |
|---|---|---|---|---|---|---|---|---|---|---|
| | | val | testA | testB | val | testA | testB | val | test | gIoU |
| X-Decoder (L) [229] | VGM | - | - | - | - | - | - | 64.6 | - | - |
| SEEM (L) [230] | | - | - | - | - | - | - | 65.6 | - | - |
| UNINEXT (R50) [195] | | 77.9 | 79.7 | 75.8 | 66.2 | 71.2 | 59.0 | 70.0 | 70.5 | - |
| UNINEXT (H) [195] | | 82.2 | 83.4 | 81.3 | 72.5 | 76.4 | 66.2 | 74.7 | 76.4 | - |
| GLEE-Pro [190] | | 80.0 | - | - | 69.6 | - | - | 72.9 | - | - |
| LISA-7B [89] | MLLM | 74.1 | 76.5 | 71.1 | 62.4 | 67.4 | 56.5 | 66.4 | 68.5 | 44.4 |
| LISA-7B* [89] | | 74.9 | 79.1 | 72.3 | 65.1 | 70.8 | 58.1 | 67.9 | 70.6 | 52.9 |
| PixelLM [151] | | 73.0 | 76.5 | 68.2 | 66.3 | 71.7 | 58.3 | 69.3 | 70.5 | - |
| PixelLLM [193] | | 76.9 | 78.5 | 74.4 | 69.2 | 72.1 | 64.5 | 70.7 | 72.4 | - |
| AnyRef* [63] | | 76.9 | 79.9 | 74.2 | 70.3 | 73.5 | 61.8 | 70.0 | 70.7 | - |
| GROUNDINGHOG [217] | | 78.5 | 79.9 | 75.7 | 70.5 | 75.0 | 64.9 | 74.1 | 74.6 | 56.2 |
| GLaMM [148] | | 79.5 | 83.2 | 76.9 | 72.6 | 78.7 | 64.6 | 74.2 | 74.9 | - |
| VisionLLM v2 | | 76.6 | 79.3 | 74.3 | 64.5 | 69.8 | 61.5 | 70.7 | 71.2 | 51.0 |

Table A3: **Comparison of referring expression segmentation performance.** gIoU denotes the general IoU. The results for RefCOCO/+/g [204, 126] are reported based on cumulative IoU (cIoU). VGM and MLLM represent vision generalist models and multimodal large language models, respectively. *The model is finetuned on the dataset.

## A.2 Evaluation on Various Domains.

**Salient Object Detection.** We compare the results of VisionLLM v2 with state-of-the-art methods for salient object detection (SOD) in Table A6. Our model clearly achieves the highest performance on 4 of the 5 classical benchmarks, demonstrating its strong object discovery capabilities.

**Camouflaged Object Detection.** The performance comparisons for camouflaged object detection (COD) are presented in Table A7. It is observed that VisionLLM v2 exhibits competitive performance with state-of-the-art expert models that undergo longer training schedule, *e.g.*, 150 epochs.

| method | backbone | iters | mIoU |
|---|---|---|---|
| Mask2Former (T)* [35] | Swin-T | | 47.7 |
| X-Decoder (T)* [229] | Focal-T | 160k | 51.0 |
| OpenSeeD (T)* [211] | Swin-T | | 52.2 |
| VisionLLM v2 | Swin-T | - | 38.9 |
| VisionLLM v2 * | Swin-T | 45k | 52.3 |

Table A4: **Comparison of semantic segmentation performance on ADE20K.** *The model is finetuned on the dataset.

| method | Point | | Scribble | | Box | | Mask | |
|---|---|---|---|---|---|---|---|---|
| | mIoU | cIoU | mIoU | cIoU | mIoU | cIoU | mIoU | cIoU |
| SAM-B [82] | 48.7 | 33.6 | - | - | 73.7 | 68.7 | - | - |
| SAM-L [82] | 51.8 | 37.7 | - | - | 76.6 | 71.6 | - | - |
| SEEM-B [230] | 47.8 | 57.8 | 43.0 | 44.0 | 44.9 | 42.1 | 48.4 | 65.0 |
| PSALM [218] | 64.3 | 74.0 | 66.9 | 80.0 | 67.3 | 80.9 | 67.6 | 82.4 |
| VisionLLM v2 | 49.1 | 60.7 | 54.7 | 72.3 | 59.1 | 78.2 | 59.6 | 81.0 |
| VisionLLM v2 * | 65.4 | 70.9 | 66.8 | 77.2 | 74.2 | 83.2 | 67.9 | 83.8 |

Table A5: **Comparison of interactive segmentation performance.** The task is evaluated on the COCO-interactive dataset proposed by [218]. *The model is finetuned on the task.

| method | DUTS | | | | DUT-OMRON | | | | HKU-IS | | | | ECSSD | | | | PASCAl-S | | | |
|---|---|---|---|---|---|---|---|---|---|---|---|---|---|---|---|---|---|---|---|---|
| | $S_m \uparrow$ | $E_m \uparrow$ | $F_\beta^\omega \uparrow$ | $\mathcal{M} \downarrow$ | $S_m \uparrow$ | $E_m \uparrow$ | $F_\beta^\omega \uparrow$ | $\mathcal{M} \downarrow$ | $S_m \uparrow$ | $E_m \uparrow$ | $F_\beta^\omega \uparrow$ | $\mathcal{M} \downarrow$ | $S_m \uparrow$ | $E_m \uparrow$ | $F_\beta^\omega \uparrow$ | $\mathcal{M} \downarrow$ | $S_m \uparrow$ | $E_m \uparrow$ | $F_\beta^\omega \uparrow$ | $\mathcal{M} \downarrow$ |
| PoolNet [109] | .878 | .889 | .880 | .040 | .828 | .863 | .808 | .056 | .910 | .949 | .933 | .032 | .922 | .924 | .944 | .039 | .847 | .850 | .869 | .074 |
| LDF [186] | .892 | .910 | .898 | .034 | .838 | .873 | .820 | .051 | .919 | .954 | .939 | .027 | .924 | .925 | .950 | .034 | .856 | .865 | .874 | .059 |
| VST [110] | .896 | .892 | .890 | .037 | .850 | .861 | .825 | .058 | .928 | .953 | .942 | .029 | .932 | .918 | .951 | .033 | .865 | .837 | .875 | .061 |
| SelfReformer [208] | .911 | .920 | .916 | .026 | .856 | .886 | .836 | .041 | .930 | .959 | .947 | .024 | .935 | .928 | .957 | .027 | .874 | .872 | .894 | .050 |
| BBRF [124] | .908 | .927 | .916 | .025 | .855 | .887 | .843 | .042 | .935 | .965 | .958 | .020 | .939 | .934 | .963 | .022 | .871 | .867 | .891 | .049 |
| EVPv2 [113] | .915 | .948 | .923 | .027 | .862 | .895 | .857 | .047 | .932 | .963 | .953 | .023 | .935 | .957 | .958 | .028 | .879 | .917 | .869 | .053 |
| VisionLLM v2 | .921 | .955 | .911 | .024 | .882 | .920 | .846 | .041 | .941 | .975 | .946 | .016 | .950 | .974 | .959 | .018 | .892 | .933 | .877 | .044 |

Table A6: **Comparison of salient object detection performance.** The metrics include S-measure ($S_m$), weighted F-measure ($F_\beta^\omega$), E-measure($E_m$) and mean absolute error ($\mathcal{M}$).

| method | CAMO | | | | COD10K | | | |
|---|---|---|---|---|---|---|---|---|
| | $S_m \uparrow$ | $F_\beta^\omega \uparrow$ | $E_m \uparrow$ | $\mathcal{M} \downarrow$ | $S_m \uparrow$ | $F_\beta^\omega \uparrow$ | $E_m \uparrow$ | $\mathcal{M} \downarrow$ |
| ZoomNet [137] | .820 | .752 | .878 | .066 | .838 | .729 | .888 | .029 |
| HitNet [67] | .849 | .809 | .906 | .055 | .871 | .806 | .935 | .023 |
| FSPNet [70] | .856 | .799 | .899 | .050 | .851 | .735 | .895 | .026 |
| ZoomNeXt [138] | .889 | .857 | .945 | .041 | .898 | .827 | .956 | .018 |
| VisionLLM v2 | .856 | .829 | .914 | .057 | .877 | .818 | .934 | .024 |

Table A7: **Comparison of camouflaged object detection performance.** The evaluation metrics including S-measure ($S_m$), weighted F-measure ($F_\beta^\omega$), E-measure($E_m$) and mean absolute error ($\mathcal{M}$).

**Visualization across various domains.** Besides the quantitative results, we also display the visualization results of VisionLLM v2 across various domains. As illustrated in Figure A1, our model also shows strong perception capacities for remote sensing, PCB, and medical images.

## A.3 Zero-shot Evaluation and In-Context Evaluation

**Zero-shot Image Captioning.** Benefiting from the joint training on large-scale vision-language datasets, VisionLLM v2 exhibits promising capacities for zero-shot image captioning. As shown in Table A1a, both VisionLLM v2-Chat and VisionLLM v2 achieve competitive performance on Flickr30K [142] and NoCaps [6] compared with previous methods.

**Zero-shot Object Detection on OdinW13.** We conduct the zero-shot object detection evaluation on OdinW13 dataset [98], as shown in Table A8. The results demonstrate that our VisionLLM v2 with a Swin-Tiny backbone is even on par with GLEE-Plus [190] with a Swin-Large backbone in $AP_{avg}$. This indicates that our model benefits from the extensive dataset joint training, thereby providing robust general object detection capabilities.

**In-Context Segmentation & In-Context Image Captioning.** To evaluate the in-context learning ability of VisionLLM v2, we compare the results of in-context segmentation and in-context image captioning in Table A9. For in-context segmentation, we construct a benchmark based on the validation set of COCO2017, where the number of in-context examples used during inference ranges from 1 to 5. For in-context image-captioning, we follow the same evaluation protocol as OpenFlamingo [11] and use 4-shot to assess the performance between different methods. The validation set is built upon COCO2017. From the table, VisionLLM v2 exhibits clear performance advantages compared with state-of-the-art methods in both in-context learning settings, which demonstrates the superior in-context capacities of our method.

| *In-Context Segmentation* | |
|---|---|
| Method | mIoU |
| Painter [183] | 44.26 |
| SegGPT [184] | 54.25 |
| VisionLLM v2 | 67.51 |
| *In-Context Image Captioning* | |
| Method | METEOR / CIDEr |
| OpenFlamingo [11] | 13.80 / 104.61 |
| VisionLLM v2 | 18.56 / 152.74 |

Table A9: Comparison of in-context segmentation and in-context image captioning performance.

## A.4 More Ablation Studies

**Super-Link Queries v.s. Token Embeddings in LISA [89].** Current MLLMs [89, 148, 151] introduce a segmentation token [SEG] into the LLM vocabulary and directly use its corresponding token embedding as a condition for SAM [82] to achieve pixel-level segmentation, which we refer to as the token embedding method. We also ablate this method for linking the LLM with task-specific decoders, as shown in Table A10. The performance difference between the two methods is negligible for tasks using text prompts, such as instance segmentation. We hypothesize that this is because the category names are seen during training, allowing the token embeddings to

| Method | Backbone | PascalVOC | AerialDrone | Aquarium | Rabbits | EgoHands | Mushrooms | Packages | Raccoon | Shellfish | Vehicles | Pistols | Pothole | Thermal | $AP_{avg}$ |
|---|---|---|---|---|---|---|---|---|---|---|---|---|---|---|---|
| GLIP-T | Swin-T | 56.2 | 12.5 | 18.4 | 70.2 | 50.0 | 73.8 | 72.3 | 57.8 | 26.3 | 56.0 | 49.6 | 17.7 | 44.1 | 46.5 |
| GLEE-Lite | ResNet50 | 61.7 | 7.9 | 23.2 | 72.6 | 41.9 | 51.6 | 32.9 | 51.1 | 35.0 | 59.4 | 45.6 | 21.8 | 56.9 | 43.2 |
| GLEE-Plus | Swin-L | 67.8 | 10.8 | 38.3 | 76.1 | 47.4 | 19.2 | 29.4 | 63.8 | 66.7 | 63.8 | 62.6 | 15.3 | 66.5 | 48.3 |
| VisionLLM v2 | Swin-T | 54.2 | 16.5 | 27.0 | 79.6 | 44.7 | 29.0 | 64.7 | 54.2 | 49.7 | 61.2 | 64.8 | 14.6 | 57.1 | 48.3 |

Table A8: **Comparison of zero-shot object detection performance on OdinW13.**

| method | query/token number | inst seg. | | ground. | pose | interact seg. | |
|---|---|---|---|---|---|---|---|
| | | $AP_b$ | $AP_m$ | P@.5 | AP | mIoU | cIoU |
| super-link queries | 1 | 50.4 | 39.6 | 85.8 | 43.0 | 43.2 | 60.0 |
| | 4 | 52.0 | 41.0 | 85.7 | 71.0 | 44.8 | 60.4 |
| | 8 | 52.1 | 40.7 | 86.4 | 71.6 | 45.9 | 61.9 |
| token embeddings | 1 | 50.8 | 39.3 | 85.4 | 42.2 | 42.1 | 57.5 |
| | 4 | 51.5 | 41.0 | 86.2 | 71.3 | 43.7 | 59.7 |
| | 8 | 52.1 | 41.1 | 86.5 | 71.5 | 44.0 | 59.1 |

Table A10: **Ablation on the comparison between super-link queries and token embeddings.** We evaluate the results on the four crucial visual perception tasks: instance segmentation, visual grounding, pose estimation and interactive segmentation. Our default setting is marked in gray .

effectively capture category semantics. However, our super-link queries method outperforms the token embedding method for more open-ended tasks, such as interactive segmentation with visual prompts, demonstrating the greater flexibility of our approach.

We emphasize two fundamental differences between the two methods: (1) The token embedding method requires sequential prediction of the special tokens during inference, which is time-consuming when the number of tokens is large. In contrast, our super-link technique requires only a single forward pass and the super-link queries would be automatically appended. This is efficient for cases requiring many tokens, such as image generation. (2) The super-link queries are not constrained by the cross-entropy loss of the LLM, allowing for more flexible and stronger representations for open-ended tasks.

### A.5 Qualitative Results

**Visual Perception.** We evaluate VisionLLM v2 on various visual perception tasks and display the visualization results from Figure A2 to Figure A6. The qualitative examples showcase that VisionLLM v2 exhibits strong visual perception capacities, from coarse to fine-grained perception (box, keypoint, pixel), from basic to novel classes, from commonly-seen domains to long-tailed domains (natural scenes, industry, agriculture, *etc*).

**Visual Generation.** Figure A7 shows more text-to-image generation results of VisionLLM v2. It could be observed that our model could generate high-quality images that not only properly follow the concepts and relations but also different styles specified in the instructions. Figure A8 shows more instructed-based image editing results of VisionLLM v2. Our model could successfully perform image editing for over five types of editing instructions, such as style transfer, object replacement, object addition, and attribute change.

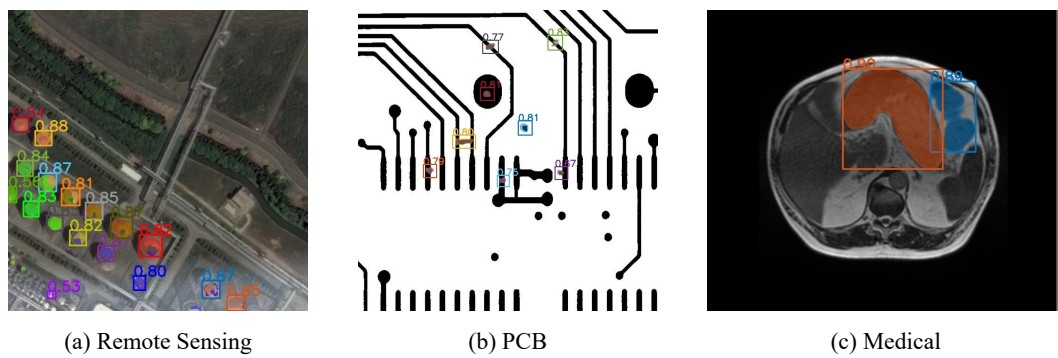

(a) Remote Sensing      (b) PCB      (c) Medical

Figure A1: **Visualization results across various domains.** VisionLLM v2 shows a strong generalization ability for remote sensing, PCB, and medical images.

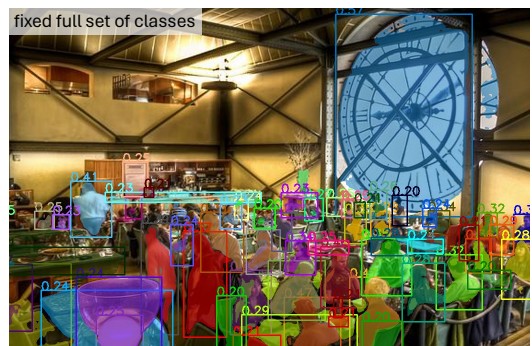

*Please conduct object detection to any [List of COCO classes] that may be present.*

*Please conduct object detection to any [List of COCO classes] that may be present.*

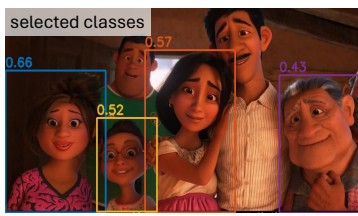

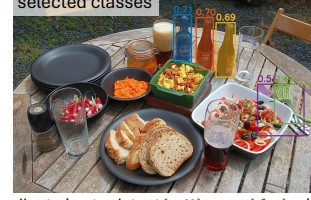

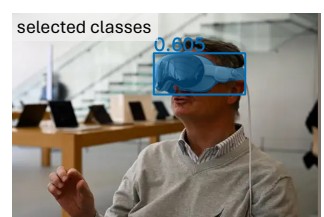

*Can you carry out object detection on this image and identify the women it contains?*

*I'm trying to detect bottles and forks in the image. Can you help me?*

*Are you capable of identifying Apple Vision Pro within this image?*

Figure A2: **Object detection and instance segmentation.** The model excels in various environments, supporting the detection of a large number of instances. Its flexibility is highlighted by its ability to detect only user-selected categories and identify novel classes.

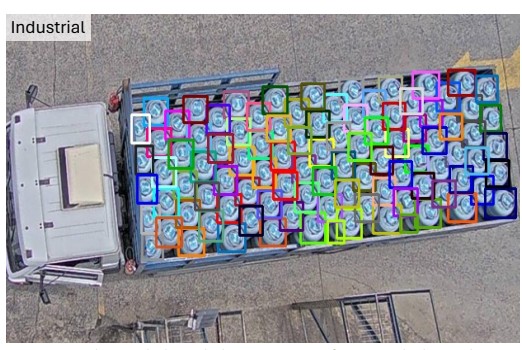

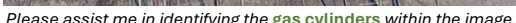

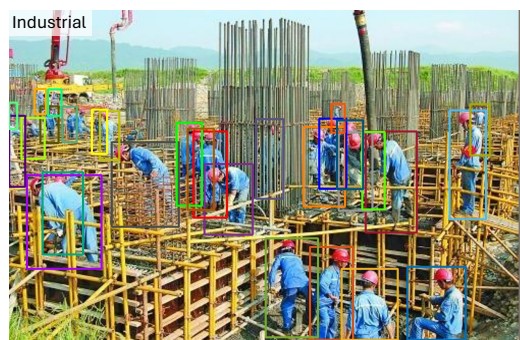

*Please assist me in identifying the gas cylinders within the image.*

*Please assist me in identifying the workers within the image.*

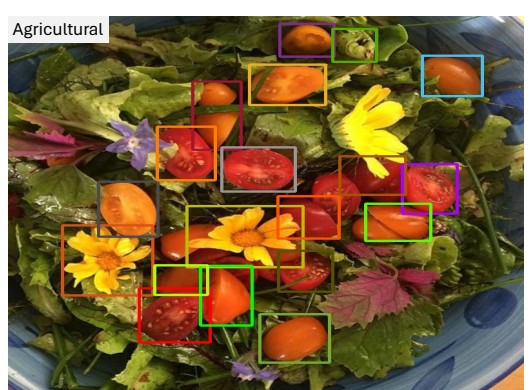

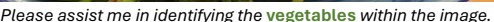

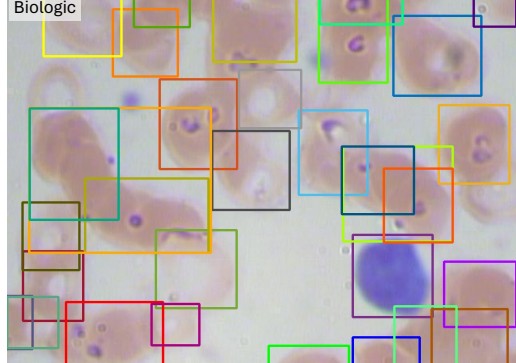

*Please assist me in identifying the vegetables within the image.*

*Please assist me in identifying the red blood cells within the image.*

Figure A3: **Object detection on multiple domains.** The image illustrates the domain adaptability of our model, which supports perception across multiple fields such as industrial, agricultural, and biological environments.

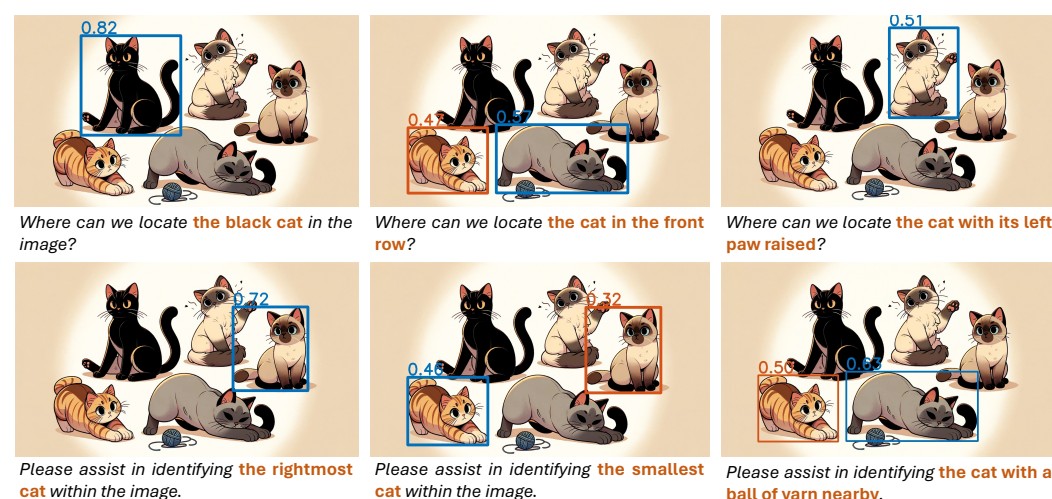

*Where can we locate **the black cat** in the image?*

*Where can we locate **the cat in the front row**?*

*Where can we locate **the cat with its left paw raised**?*

*Please assist in identifying **the rightmost cat** within the image.*

*Please assist in identifying **the smallest cat** within the image.*

*Please assist in identifying **the cat with a ball of yarn nearby**.*

Figure A4: **Visual grounding.** On the visual grounding task, our model demonstrates good accuracy and a certain level of reasoning capability.

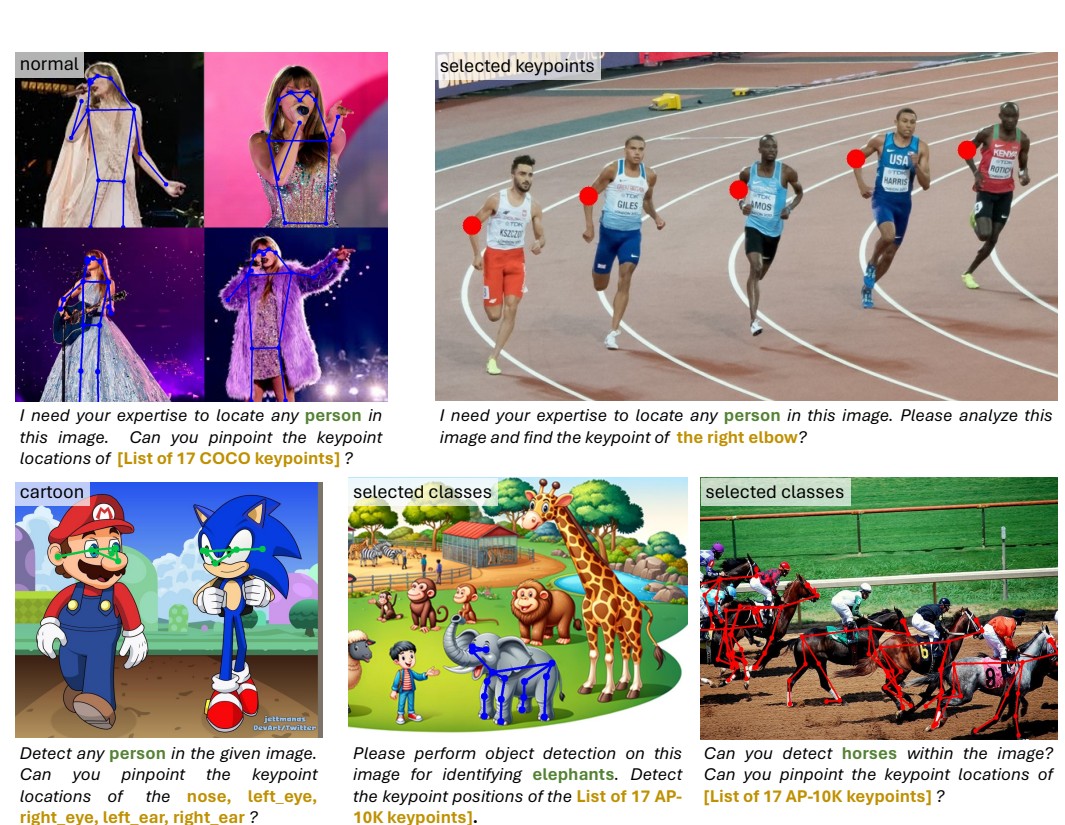

*I need your expertise to locate any **person** in this image. Can you pinpoint the keypoint locations of **[List of 17 COCO keypoints]** ?*

*I need your expertise to locate any **person** in this image. Please analyze this image and find the keypoint of **the right elbow**?*

*Detect any **person** in the given image. Can you pinpoint the keypoint locations of the **nose, left_eye, right_eye, left_ear, right_ear** ?*

*Please perform object detection on this image for identifying **elephants**. Detect the keypoint positions of the **List of 17 AP-10K keypoints**.*

*Can you detect **horses** within the image? Can you pinpoint the keypoint locations of **[List of 17 AP-10K keypoints]** ?*

Figure A5: **Pose estimation.** Our model is capable of detecting keypoints in humans and animals with flexibility. The model allows users to select specific instance categories for detection, as well as choose individual keypoints.

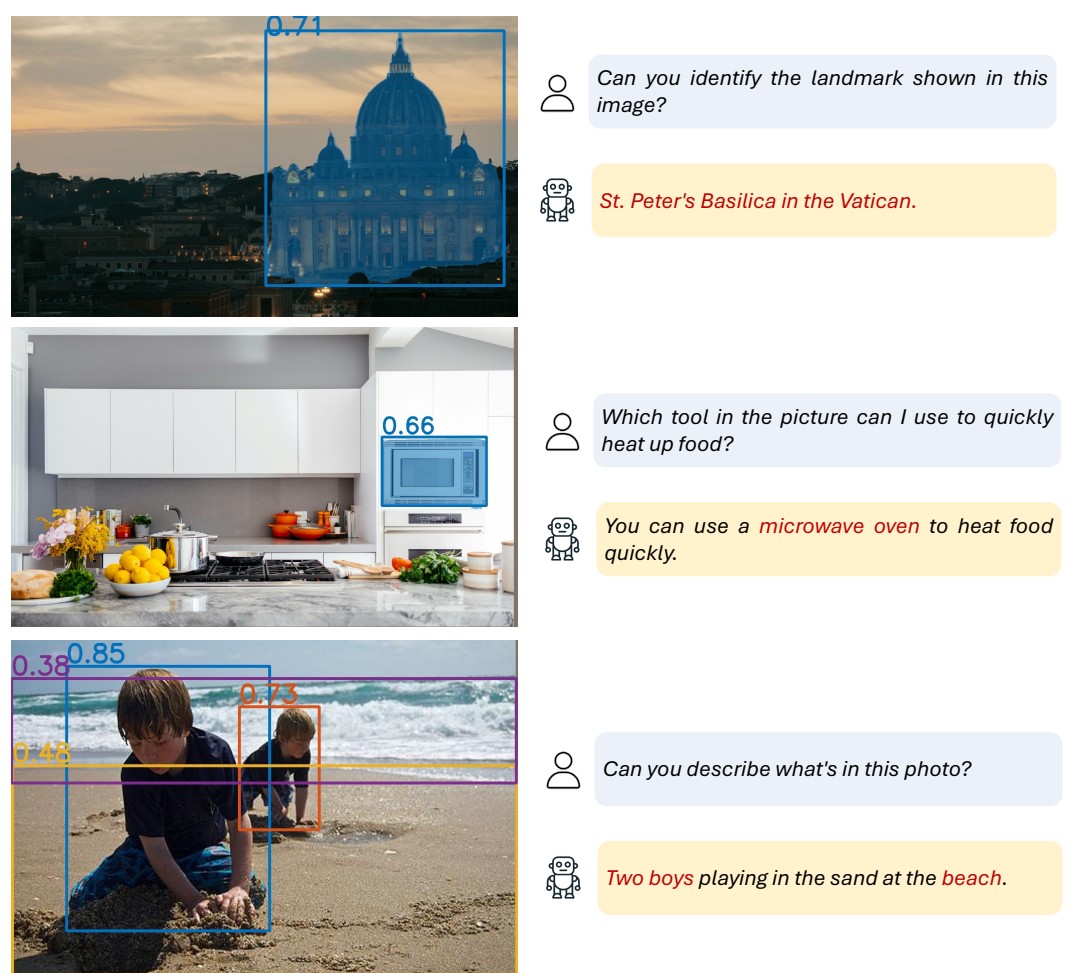

Figure A6: **Grounded caption.** The model accurately locates objects based on user prompts, outputs bounding boxes, and provides answers to user queries.

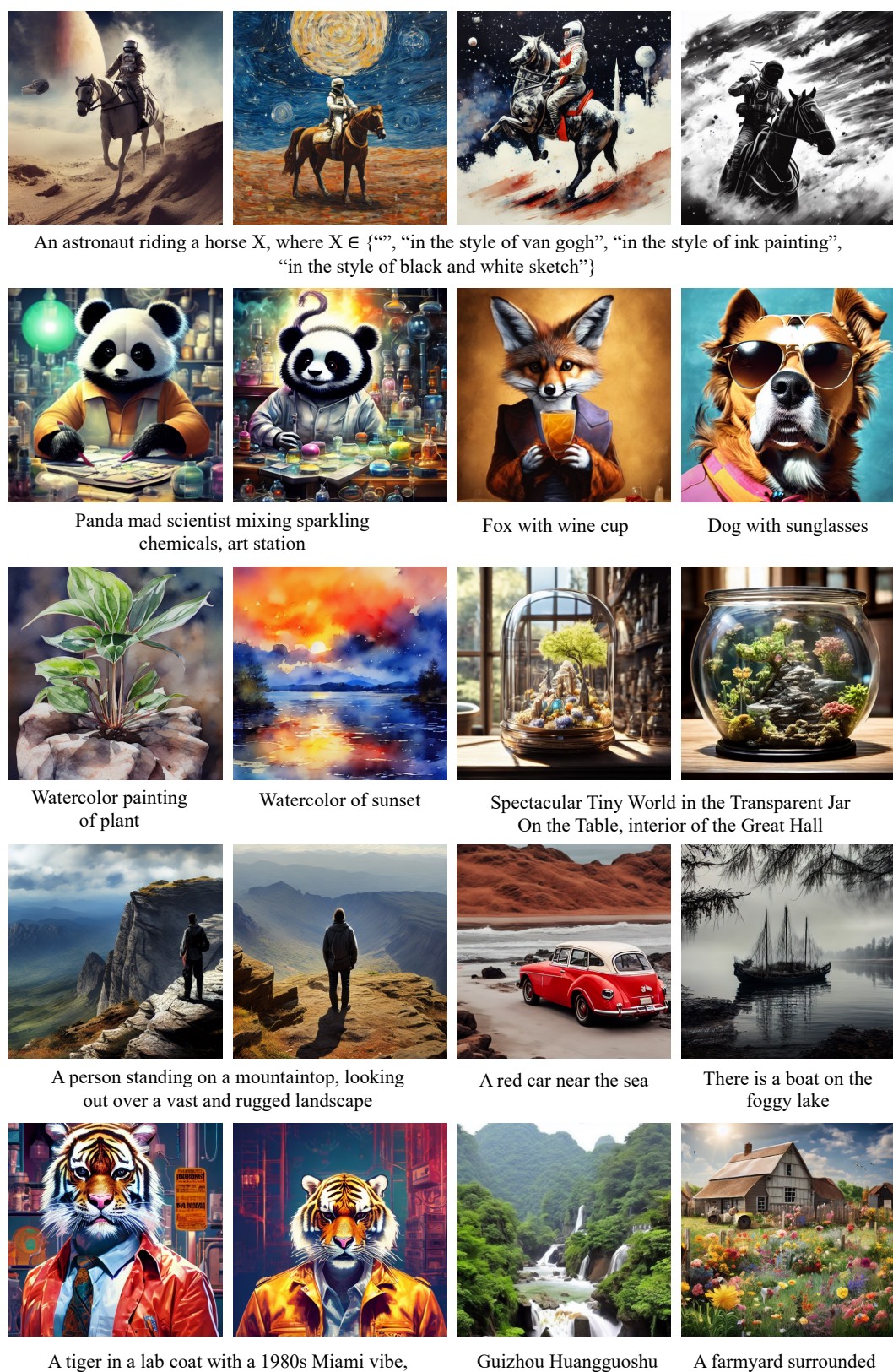

An astronaut riding a horse X, where X ∈ {"", "in the style of van gogh", "in the style of ink painting", "in the style of black and white sketch"}

Panda mad scientist mixing sparkling chemicals, art station

Fox with wine cup

Dog with sunglasses

Watercolor painting of plant

Watercolor of sunset

Spectacular Tiny World in the Transparent Jar On the Table, interior of the Great Hall

A person standing on a mountaintop, looking out over a vast and rugged landscape

A red car near the sea

There is a boat on the foggy lake

A tiger in a lab coat with a 1980s Miami vibe, digital art

Guizhou Huangguoshu Waterfall

A farmyard surrounded by beautiful flowers

Figure A7: **VisionLLM v2 text-to-image generation examples.** VisionLLM v2 could generate high-quality images that not only properly follow the concepts and relations, but also different styles specified in the instructions. .

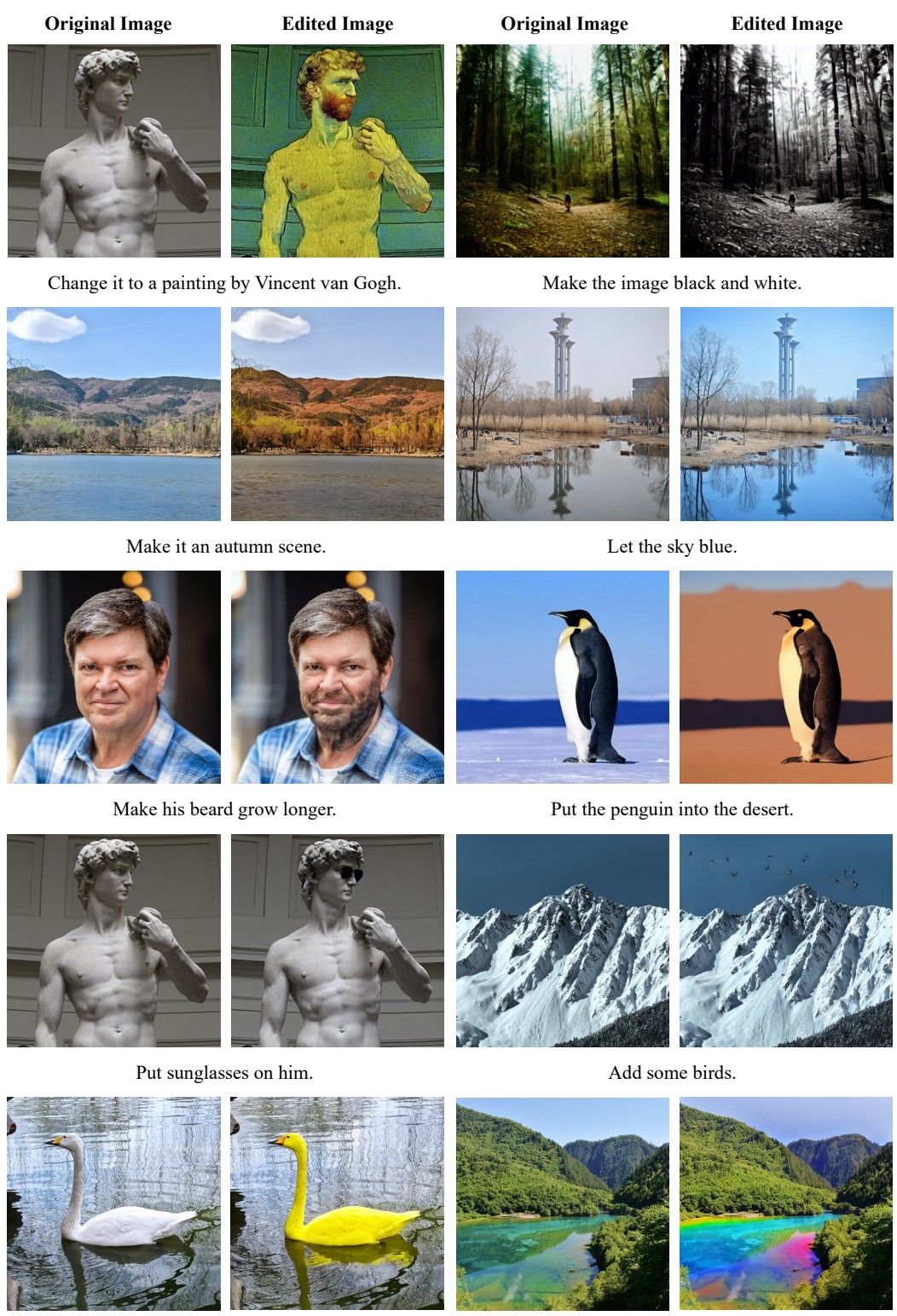

| Original Image | Edited Image | Original Image | Edited Image |
| --- | --- | --- | --- |

Change it to a painting by Vincent van Gogh.          Make the image black and white.

Make it an autumn scene.          Let the sky blue.

Make his beard grow longer.          Put the penguin into the desert.

Put sunglasses on him.          Add some birds.

Turn the goose yellow.          Make the river a rainbow

Figure A8: **VisionLLM v2 instructed-based image editing examples.** VisionLLM v2 can understand a variety of instructions such as style transfer, object replacement, object addition, attribute change, and more to generate high-quality edited images.

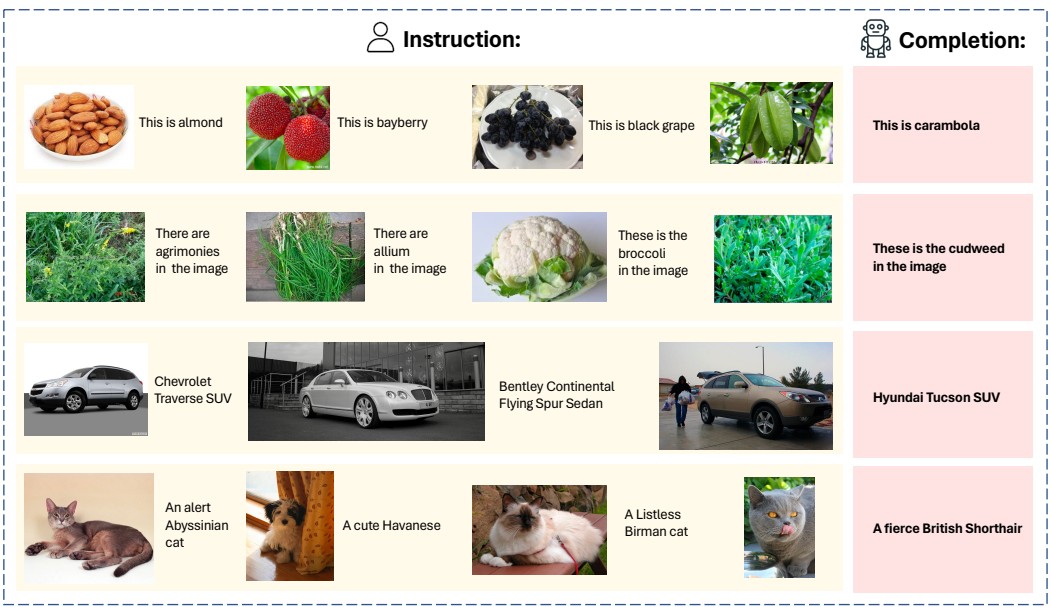

Figure A9: **In-context fine-grained visual recognition.** It demonstrates that our model has the strong capability of fine-grained recognition.

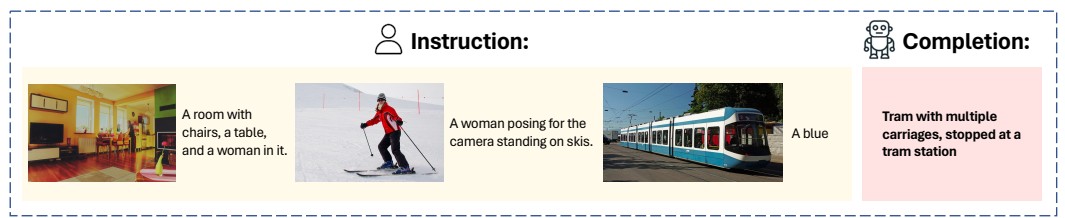

Figure A10: **In-context image captioning.** Our model is able to perform text completion based on in-context examples.

**Multimodal In-context Learning Ability.** To qualitatively verify the in-context capabilities of our model after trained on MMIC, we provide comprehensive visualizations across different tasks. As demonstrated in Figure A9, A10, A11 and A12, our method can handle both visual and textual prompts, enabling it to perform tasks that require understanding and integration of information from different modalities. In addition, our models can distinguish between different prompting strategies and can correctly use the corresponding detection or segmentation tools to obtain the expected output based on given in-context examples.

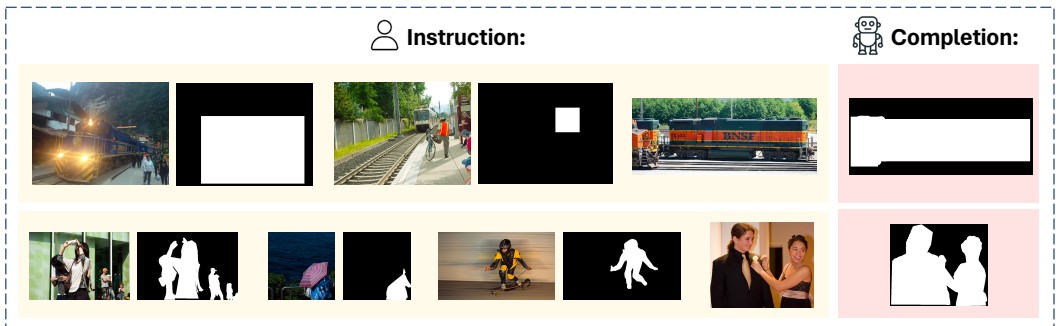

Figure A11: **In-context detection and segmentation.** We just need to provide some examples where the instances falling into the same class are highlighted. Then our model can learn from the example and use the too of detection or segmentation to process the input image.

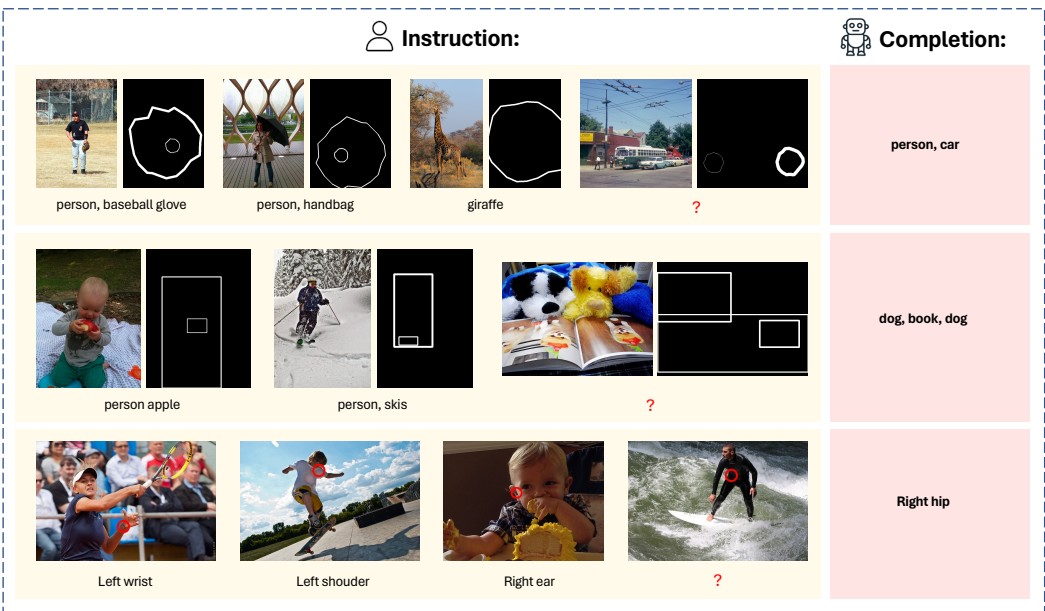

Figure A12: **In-context regional perception.** In our dataset, we construct various visual masks in input prompts. Our models are required to infer from the given examples and complete the text for the last image.

# B    More Architecture Details

## B.1    Region Encoder

The region encoder is designed to encode various shaped visual prompts such as points, scribbles, boxes, *etc*. Each visual prompt is represented by a binary mask. We first concatenate the binary mask with the image along the channel dimension, resulting in a 4-channel input, denoted as $I_{\text{vprt}} \in \mathbb{R}^{4 \times H \times W}$. The region encoder is implemented with three convolutional layers: the first layer uses a kernel size of 7 and a stride of 7, the second layer employs a kernel size of 2, and a stride of 2, and the final layer features a kernel size of 1 and a stride of 1. Each convolutional layer is followed by layer normalization [12] and GELU activation [65]. This process downsamples the input $I_{\text{vprt}} \in \mathbb{R}^{4 \times H \times W}$ by a factor of 14. We further augment this feature map by adding the feature map of the global image $I_{\text{global}}$. Finally, we use grid sampling to extract features within the masked regions and pool them into a single region embedding $F_{\text{vprt}} \in \mathbb{R}^{1 \times C}$.

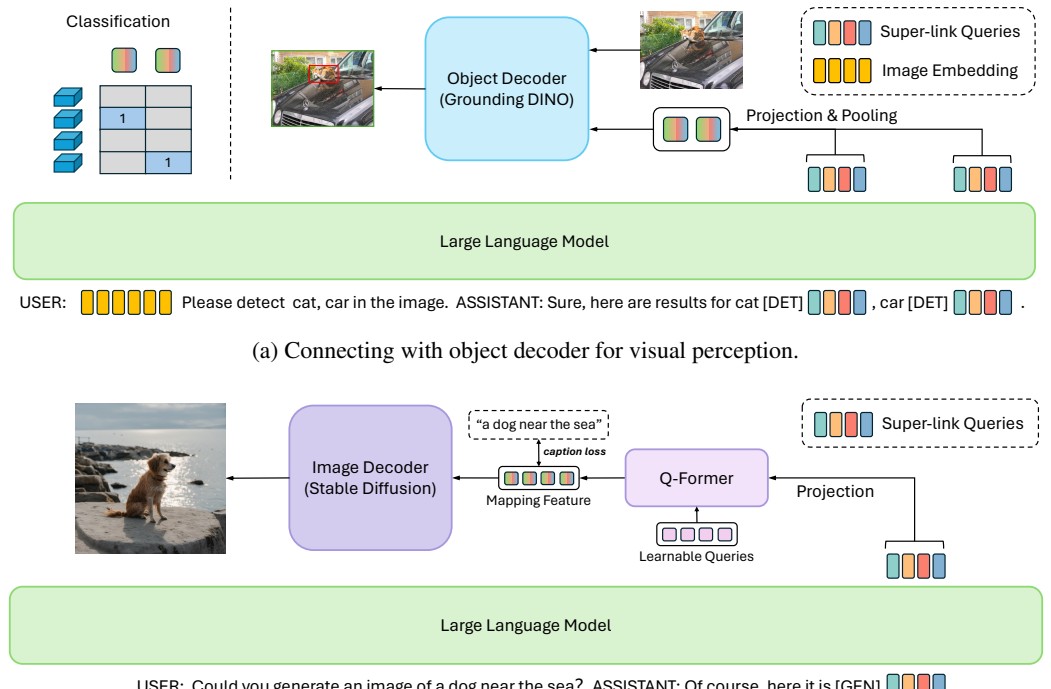

(a) Connecting with object decoder for visual perception.

(b) Connecting with image decoder for visual generation.

Figure A13: **Architecture details for connecting LLM with task-specific decoder via super-link queries.** (a) Connecting with object decoder. We first extract the per-category features by performing projection and pooling on the hidden states of corresponding super-link queries. Then these features are sent into the object decoder as text features. (b) Connecting with image decoder. We add a Q-Former for projecting the features of super-link queries to the feature space of Stable Diffusion.

## B.2   Task-specific Decoders

In this subsection, we provide more explanations about how to connect LLM with task-specific decoders via super-link queries, which enables the end-to-end optimization of the entire network.

**Connecting with Object Decoder.** For visual perception tasks like object detection, we employ Grounding DINO [112] as the object decoder to localize objects as well as classify their categories. To achieve this, LLM would output each category name in the response, followed by a special token [DET] and super-link queries. We then obtain the per-category features by extracting the hidden states of LLM for corresponding super-link queries and pooling them into one embedding. Grounding DINO receives both the image and the obtained per-category features as inputs and predicts the detection results. The process is illustrated in Figure A13a. It is noted that we discard the text encoder in the original Ground DINO and use the obtained per-category features as text features to perform the vision-language alignment for classification. During training, the total loss includes the cross-entropy loss of LLM and detection loss of the object decoder. Similarly, the keypoint decoder is also integrated into the LLM in the same way and performs pose estimation.

**Connecting with Image Decoder.** We utilize Stable Diffusion [152] as the image decoder and take the example of text-to-image generation for clarification, as depicted in Figure A13b. The super-link queries are appended after the special token [GEN] in the LLM's response. After passing through the LLM, an MLP layer and a lightweight Q-Former [97, 83] module are added to project the features of the super-link queries into the representation space of Stable Diffusion, i.e., mapping features. We bypass the text encoder in Stable Diffusion and directly use the mapping features as the text embedding condition. During training, in addition to the next token loss in the LLM, we employ two MSE losses for supervision: one between the encoded text features by CLIP [144] and the mapping features, and the other between the ground-truth images/noise and predicted images/noise.

| task | #sample | dataset |
|---|---|---|
| Conversation | 1.59M | ShareGPT4V [28], Laion-GPT4V [1], ALLaVA [25] |
| Image Captioning | 0.59M | COCO [31], TextCaps [159] |
| Image VQA | 5.19M | ShareGPT4V [28], GRIT [140], VQAv2 [57], OK-VQA [127], A-OKVQA [127], GQA [71], AI2D [78], ScienceQA [154] |
| OCR | 0.58M | OCR-VQA [130], ChartQA [128], DocQA [40], STVQA [18], DVQA [75], InfoVQA [129] LLaVAR [216], GeoQA+ [23], SynthDoG [81] |
| Region Captioning | 2.66M | Visual Genome [86], RefCOCO/+/g [204, 126], Flickr30K [142], All-Seeing [181] |
| Region VQA | 1.80M | VCR [209], Osprey [207], All-Seeing [181] |
| Region Recognition | 0.40M | V3Det [177], COCO [104], LVIS [61] |

(a) Datasets used in stage-1.

| task | #sample | dataset |
|---|---|---|
| Object Detection & Instance Segmentation | 1.18M | COCO [104], LVIS [61], Objects365 [155], OpenImages [87], CrowdHuman [156], NC4K [122], COD10K [46], CAMO [90], CPD1K [220], DUTS [179], MSRA10K [36], DOTA [192], SARDet-100K [102], DeepPCB [167] |
| Grounded Caption | 0.18M | Flickr30K [142], Groma-Instruct [123] |
| Semantic Segmentation | 0.13M | ADE20K [222], CityScapes [41], Mapillary [132], LoveDA [178], Medical MRI [200] |
| Interactive Segmentation | 0.34M | COCO [104], SA-1B [82] |
| Visual Grounding | 0.13M | RefCOCO/+/g [204, 126], ReasonSeg [89] |
| Pose Estimation | 0.25M | COCO [104], CrowdPose [96], Human-Art [74], AP-10K [203], APT-36K [199], MacaquePose [88], 300W-Face [153], Animal Kingdom [133], AnimalWeb [79], Vinegar Fly [141], Desert Locust [58] |
| Object Counting | 0.61M | COCO [104], LVIS [61], Objects365 [155], OpenImages [87], CrowdHuman [156] CA44 [73], HierText [118] |
| Image Generation & Edit | 5.90M | JourneyDB [135], LAION-Aesthetics [3], InstructPix2Pix [20] |
| Multimodal In-Context | 0.89M | MMIC (ours) |

(b) Datasets used in stage-3.

Table A11: **Summary of datasets used in each training stage.** The datasets used in stage-2 is the combination of stage-1 and stage-3 datasets, which enables the model to learn multiple capacities without comprising its conversation ability. For some large-scale datasets such as SA-1B [82], we randomly sample a subset from them for training.

| prompt pattern | Task | #sample | dataset |
|---|---|---|---|
| $IMT \rightarrow T$ | VQA with visual marks | 147K | COCO [104], TextOCR [161], AP10K [203] ICDAR2019 [165], AP10K [203] |
| $[IT]I \rightarrow T$ | In-context VQA, In-context captioning, In-context visual recognition | 465K | Food-101 [19], Oxford Flower [134], CUB-200-2011 [175], Stanford Dogs [80], Oxford-IIIT Pet [139], Stanford Cars [85], Birdsnap [16], VegFru [66], iNaturalist 2021 [174], UECFOOD-256 [77], CNFOOD-241 [45], ALLaVA [25], COCO [104] |
| $[IMT]IM \rightarrow T$ | In-context visual recognition with visual marks | 40K | COCO [104], TextOCR [161], ICDAR2019 [165], AP10K [203] |
| $[IM]I \rightarrow M$ | In-context object detection In-context segmentation In-context OCR | 240K | COCO [104], TextOCR [161], ICDAR2019 [165] |
| Total | | 892K | |

Table A12: **Datasets used for visual prompting tasks and in-context visual tasks.** In the table, *I* denotes Image, *M* denotes Mask, such as segmentation mask or visual prompts, and *T* denotes Text. In addition, we use "[*]" to represent that the item within "[]" repeats one or more times.

## C    More Dataset Details

To support the training for enhancing our model with various capacities, we meticulously collect and re-organize the datasets from a broad range of tasks. These data are publicly available, and we comprehensively list all the data we used in Table A11. In addition to the commonly used dataset for the standard vision and vision-language tasks, we find that many works explore visual prompting strategies and in-context learning. However, there is still a lack of public datasets focusing on addressing these tasks currently. To this end, we organize a series of datasets into a new one coined as a multimodal in-context (**MMIC**) dataset to facilitate the model with in-context learning abilities, applicable to both visual and textual prompts. As shown in Table A12, built upon several datasets, we support lots of visual prompting and in-context tasks for fine-grained visual recognition,

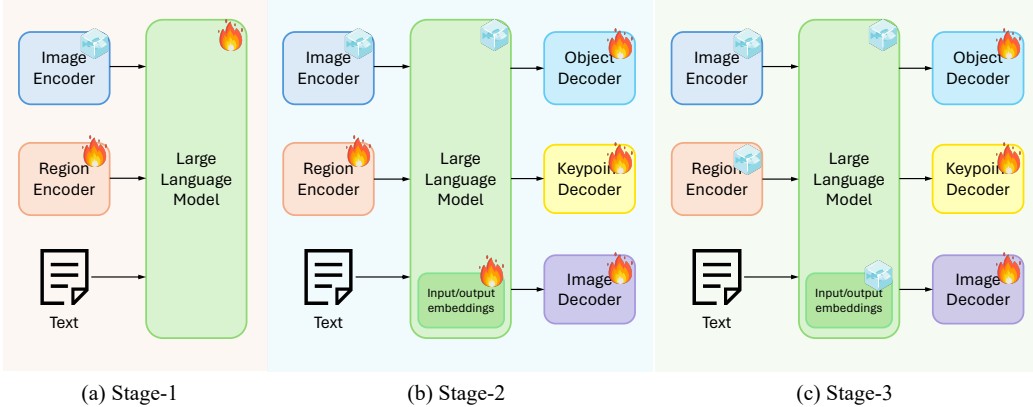

|     (a) Stage-1     |     (b) Stage-2     |     (c) Stage-3     |

Figure A14: **The training strategy of VisionLLM v2.** It consists of three consecutive stages: (1) multimodal training; (2) multi-capacity fine-tuning; (3) decoder-only fine-tuning. Leveraging this training strategy, VisionLLM v2 progressively learns the global knowledge and enhances its capacities from a broad range of data sources.

including categories such as cats, dogs, fruits, vegetables, food, cars, birds, etc. Additionally, we also make efforts on in-context object detection, in-context object segmentation, in-context captioning, in-context OCR, and in-context VAQ.

## C.1  MMIC Dataset Construction

For tasks that require visual or textual in-context examples, we randomly select $N$ samples, where $N \in [2, 6]$, without replacement from the dataset. The first $N - 1$ samples are presented as in-context examples of the model. These examples serve to provide a reference or a guide for the type of output expected. The model is then tasked with solving or completing the task based on the last sample in the sequence. This paradigm allows the model to learn from examples and apply that knowledge to new, unseen data.

Inspired by [157], visual marks can also serve as the input for multimodal LLMs. As a result, we design five types of visual marks: circle, hand-drawn circle, arrow, box, and mask. Each visual mark can be either solid or hollow. We primarily construct this dataset based on COCO [104], AP10K [203] and some OCR datasets [165, 161], where we randomly sample $M(\in [1, 5])$ instances per image. The same type of visual mark is used to highlight the selected instance within one image, ensuring consistency and clarity for the model's learning process.

The examples of constructed instructions can refer to Figure A9, A10, A11 and A12. The entire dataset has constructed a multimodal corpus with ~862K question&answer pairs. We expect that this dataset can further advance the development of this field.

## D  Training Details

Figure A14 depicts the three-stage training process. Table A13 lists the detailed training configurations of VisionLLM v2 in different training stages. In each stage, the model inherits the weights from the previous stage and continues training. The image encoder keeps frozen in all stages following previous works [107, 105].

**Settings of Stage-1.** Stage-1 consists of pretraining and instruction tuning phases as [107, 105]. As shown in Table A13, in the pretraining phase, We freeze the LLM. And only the region encoder and projections for image embedding and region embedding are trained for efficiency. We adopt the AdamW optimizer [119] with the peak learning rate of 1e-3 and weight decay of 0. The training involves a total batch size of 2048 across 64 A100 GPUs. In the instruction tuning phase, LLM is unfrozen for full-parameter training. The peak learning rate is decreased to 2.5e-5 for training stabilization. The model is trained on 64 A100 GPUs with a total batch size of 1024. And we begin adopting the dynamic resolution approach [106, 33] in this phase. The maximal number of local patches, *i.e.*, max tile, is set as 4.

| config | stage1 pretrain. | stage1 tune. | stage2 | stage3 |
|---|---|---|---|---|
| image enc. peak learning rate | frozen | frozen | frozen | frozen |
| region enc. peak learning rate | 1e-3 | 2.5e-5 | 1e-5 | frozen |
| LLM peak learning rate | frozen | 2.5e-5 | 1e-5 | frozen |
| dec. peak learning rate | - | - | 1e-4 | 1e-4 |
| learning rate schedule | cosine decay | cosine decay | cosine decay | cosine decay |
| optimizer | AdamW [119] | AdamW [119] | AdamW [119] | AdamW [119] |
| weight decay | 0. | 0. | 0. | 0. |
| input resolution | $336^2$ | $336^2$ | $336^2$ | $336^2$ |
| dynamic resolution | ✗ | ✓ | ✓ | ✓ |
| max tile | - | 4 | 4 | 4 |
| LLM LoRA rank | - | - | 32 | 32 |
| LLM LoRA alpha | - | - | 64 | 64 |
| warmup ratio | 0.03 | 0.03 | 0.03 | 0.03 |
| total batch size | 2048 | 1024 | 256 | 256 |
| epoch | 1 | 1 | 1 | 12 |
| numerical precision | DeepSpeed bf16 [149] | DeepSpeed bf16 [149] | DeepSpeed bf16 [149] | DeepSpeed bf16 [149] |
| GPUs for training | 64 × A100 (80G) | 64 × A100 (80G) | 128 × A100 (80G) | 128 × A100 (80G) |

Table A13: **Training settings of VisionLLM v2 in different stages.** Max tile means the maximal number of local patches when adopting the dynamic resolution approach [106, 33] for the images.

**Settings of Stage-2.** In stage-2, we add the task-specific decoders and perform the multi-capacity fine-tuning. In this stage, we only finetune the input and output embeddings of LLM to save computational memory and preserve the convesational ability. LLM and region encoder are trained with the peak learning rate of 1e-5, while the decoders are trained with the peak learning rate of 1e-4. The model is trained on 128 A100 GPUs with a batch size of 2 per GPU.

**Settings of Stage-3.** In stage-3, we freeze all the components except for the task-specific decoders to maintain the conversational ability. The model undergoes 12 training epochs on 128 A100 GPUs with a peak learning rate of 1e-4 and a total batch size of 256.

These three stages take around 5 / 3 / 10 days to finish the training, respectively.

**Training Losses.** During training, we use the standard cross-entropy loss in stage-1. In stage-2 and stage-3, when integrating the task-specific decoders, we simply sum the losses from the LLM and decoders directly, without reweighting each component. *i.e.*,

$$L_{\text{total}} = L_{\text{llm}} + L_{\text{gdino}} + L_{\text{unipose}} + L_{\text{sd}} + L_{\text{ip2p}} \tag{1}$$

# E    Instruction Templates

To support the proper invocation of task-specific decoders, we construct a series of instruction templates for different tasks using ChatGPT [4] and use them as instruction tuning data for LLM. We comprehensively list all the instruction templates below, from Table A14 to Table A24.

1. Can you provide a detailed description of <regions> in the image?
2. From what you see in the image, could you paint a vivid picture of what <regions> looks like?
3. What stands out to you the most about <regions> depicted in the image? Could you describe it in detail?
4. I'm interested in this image, especially in the <regions>. Can you provide a comprehensive description of it?
5. I'd like to learn about the detailed information of <regions> in this image. Can you describe its characteristics in depth?
6. The <regions> in this image seems fascinating. Can you delve into its description, highlighting its notable aspects?
7. Can you paint a vivid picture of the scenery within <regions> captured in the image?
8. Could you provide a detailed account of the environmental characteristics within <regions> in the image?
9. I'm seeking more information about <regions> in the image. Could you provide a comprehensive overview?
10. Please help me write a detailed description for <regions> in the image.

Table A14: **A list of instructions for single-region detailed caption.**

1. Can you provide me with a brief description of <regions> in the picture?
2. I'm curious about the region represented by <regions> in the picture. Could you describe it in short?
3. What can you tell me about <regions> in the image?
4. I'd like to know more about the area in the photo labeled <regions>. Can you give me a brief description?
5. Could you describe <regions> in the picture in short?
6. What content can you give me about <regions> in the photo?
7. Please provide me with a short description of <regions> in the image.
8. Can you give me a brief account of the region labeled as <regions> in the picture?
9. I'm interested in learning more about <regions> in the photo. Can you describe it in short?
10. What is the region outlined by <regions> in the picture like? Could you give me a brief description?
11. Can you provide me with a brief description of <regions> in the picture, please?
12. I'm curious about the region represented by <regions> in the picture. Could you describe it in short, please?
13. What can you tell me about <regions> in the image, exactly?
14. I'd like to know more about <regions>. Can you give me a brief description?
15. Could you describe the region shown as <regions> in the picture in short, please?
16. What content can you give me about <regions> in the photo, please?
17. Please provide me with a short description of <regions> in the image, please.
18. Can you give me a brief account of the region labeled as <regions> in the picture, please?
19. I'm interested in learning more about <regions> in the photo. Can you describe it in short, please?
20. What is <regions> in the picture like, please? Could you give me a brief description?

Table A15: **A list of instructions for single-region brief caption.**

1. Could you please give me a brief description of <regions>?
2. Can you provide a short description of <regions> in this image?
3. Please describe in short the contents of the boxed areas <regions>.
4. Could you give a brief explanation of what can be found within <regions> in the picture?
5. Could you give me a brief explanation of <regions> in this picture?
6. Can you provide a short description of <regions> in this photo?
7. Help me understand the specific locations labeled <regions> in this picture in short, please.
8. What is the brief information about the areas marked by <regions> in this image?
9. Could you provide me with a brief analysis of the regions designated <regions> in this photo?
10. What are the specific features of the areas marked <regions> in this picture that you can describe in short?
11. Could you elaborate on the regions identified by <regions> in this image?
12. What can you tell me about the areas labeled <regions> in this picture?
13. Can you provide a brief analysis of <regions> in this photo?
14. I am interested in learning more about <regions> in this image. Can you provide me with more information?
15. Could you please provide a brief description of <regions> in this photo?
16. What is the significance of the regions labeled <regions> in this picture?
17. I would like to know more about <regions> in this image. Can you provide me with more information?
18. Can you provide a brief breakdown of <regions> in this photo?
19. What specific features can you tell me about the areas identified by <regions> in this picture?
20. Could you please provide a short explanation of the locations labeled <regions> in this image?
21. Can you provide a brief account of the regions designated <regions> in this photo?
22. I am curious about the areas marked <regions> in this picture. Can you provide me with a brief analysis?
23. What important content can you tell me about the specific locations identified by <regions> in this image?
24. Could you please provide a brief description of <regions> in this photo?
25. What can you tell me about the features of the areas designated <regions> in this picture?
26. Can you provide a comprehensive overview of the regions marked <regions> in this image?
27. I would like to know more about the specific locations identified by <regions> in this photo. Can you provide me with more information?
28. What is the detailed information you have on <regions> in this picture?
29. Could you provide me with a brief analysis of <regions> in this image?
30. Can you provide a brief explanation of the specific locations marked by <regions> in this photo?

Table A16: **A list of instructions for multi-region caption.**

1. Whis is the object category of <regions>? Answer the question with a single word or phrase.
2. Could you tell me what is the object in <regions>? Answer the question with a single word or phrase.
3. What category best describes the area represented by <regions>? Answer the question with a single word or phrase.
4. Can you specify the type of object inside the region labeled by <regions>? Answer the question with a single word or phrase.
5. How would you label the area indicated by <regions> in the image? Answer the question with a single word or phrase.
6. Give a category label to the region outlined by <regions>. Answer the question with a single word or phrase.
7. Please identify the category of the object inside the <regions>. Answer the question with a single word or phrase.
8. Examine and determine the primary subject located within <regions>. Answer the question with a single word or phrase.
9. I need your help to assign an object category to the <regions>, please. Answer the question with a single word or phrase.
10. Evaluate the content of the region shown as <regions> and provide its category. Answer the question with a single word or phrase.

Table A17: **A list of instructions for region recognition.**

1. Can you analyze the image and identify the <class> present?
2. In this image, could you detect all instances of <class>?
3. Are you capable of identifying <class> within this image?
4. Could you please detect the objects you find that belong to the <class> category in the image?
5. Can you perform object detection on the image and tell me the <class> you find?
6. I'm trying to detect <class> in the image. Can you help me?
7. Can you carry out object detection on this image and identify the <class> it contains?
8. In the context of the image, I'd like to know which objects fall under the category of <class>. Is that something you can do?
9. I have an image that needs examination for objects related to <class>. Can you perform that?
10. Can you determine if there are any <class> present in the image using object detection?
11. Could you please carry out object detection on this image and list any <class> that you discover?
12. Could you help me identify the objects corresponding to <class> in the provided image?
13. Are you capable of detecting and labeling <class> objects within the image?
14. I'm curious about the objects in the image that correspond to the <class> category. Could you assist in finding them?
15. Can you detect <class> within the image and provide information about its presence?
16. Please examine the image and let me know which objects fall under the <class> category.
17. Please perform object detection on this image to identify <class>.
18. I need your expertise to locate <class> in this image.
19. Please let me know the objects falling into the <class> category in the image.
20. Please help me identify objects falling under the <class> category in this image.
21. Please assist me in identifying the <class> objects within the image.
22. Please provide a breakdown of all the <class> objects visible in the image.
23. Please analyze the image and let me know if you can find any objects categorized as <class>.
24. I'm seeking your help in identifying <class> within the contents of the image.
25. Please conduct object detection on the image to locate any <class> that may be present.
26. Please execute object detection on this image and provide details about any <class> you detect.
27. Please identify and list any <class> in the given image using object detection.
28. Please analyze the image and let me know if there are any recognizable <class> objects.
29. Detect any <class> in the given image, if possible.
30. I need assistance in recognizing the <class> shown in the image.

Table A18: **A list of instructions for object detection.**

1. Where can we locate the <expression> in the image?
2. Do you know where the <expression> is within the image?
3. Have you seen the <expression> in this image? Where is it?
4. Could you tell me where the <expression> is in the image?
5. Whereabouts in the image can we find the <expression>?
6. Do you have any idea where the <expression> might be in this image?
7. Are you aware of the <expression>'s position within the image?
8. Where in the image should we be looking for the <expression>?
9. Is it possible to identify the <expression>'s location in this image?
10. Have you figured out where the <expression> is in this image?
11. Could you provide guidance on finding the <expression> in the image?
12. Do you know where I can locate the <expression> in the picture?
13. Can you tell me the precise location of the <expression> in the image?
14. Would you be able to point out the <expression> within the image?
15. Are you able to discern the <expression> in the image?
16. Please help me locate the <expression> in the image.
17. Please find the object indicated by the expression <expression> in the image.
18. Please assist in identifying the <expression> within the image.
19. Please determine the exact position of the <expression> in the image.
20. Please ascertain the whereabouts of the <expression> in this image.
21. Please assist me in locating the <expression> within the image.
22. Please take a moment to find the object denoted by the expression <expression> in the image.
23. Please help us identify the precise location of the <expression> in this image.
24. Please provide your guidance in finding and marking the <expression> within the image.
25. Please make it a priority to discover and highlight the <expression> within the image.
26. Let's determine the specific area where the <expression> is situated in the image.
27. We're aiming to establish the spatial coordinates of the <expression> in this image.
28. We need to establish the exact whereabouts of the <expression> within the image.
29. We are actively engaged in the process of locating the <expression> in the image.
30. Let's find the <expression> within the image.

Table A19: **A list of instructions for visual grounding.**

1. Could you aid me in generating unique masks for every category present in <class> in this image?
2. Can you help me generate distinct masks for each category that belongs to <class> in this image?
3. Is it possible for you to help me create distinct masks for the different <class> categories in this image?
4. Could you assist me in generating masks that correspond to each individual <class> category in this image?
5. Would you mind helping me generate separate masks for each <class> category detected in this image?
6. Can you guide me in generating unique masks for all the categories falling under <class> in this image?
7. Can you provide me with the necessary support to generate masks specific to each <class> category in this image?
8. Could you please guide me in creating separate masks for each <class> category detected in this image?
9. Can you support me in generating masks for all the categories encompassed by <class> in this image?
10. Examine the image and generate masks that correspond to each individual <class> category present.
11. Is it possible for you to help me generate separate masks for each detected category falling under <class> in this image?
12. Can you assist me in generating masks that isolate each category belonging to <class> in this image?
13. Can you provide me with assistance in generating individual masks for every <class> category identified in this image?
14. Can you help with the process of generating masks that are specific to each <class> category detected in this image?
15. Generate masks that accurately depict each category belonging to <class> in this image.
16. I require assistance in producing separate masks for all the <class> categories in this image.
17. I need your support to generate masks that are specific to each <class> category in this image.
18. Your task is to produce masks that differentiate each category falling under the <class> category in this image.
19. Please create masks that are distinct for each category belonging to <class> in this image.
20. I'm seeking your help to generate masks that isolate every category within the <class> category in this image.
21. Please segment the different categories falling under <class> in this image and generating masks for each.
22. Please accurately segment and generate masks for all the categories falling under <class> in this image.
23. I need your support to create masks that are specific to each <class> category identified in this image.
24. I'm requesting your aid in generating masks that distinguish each category belonging to <class> in this image.
25. Please lend me your expertise in creating masks that are unique for each detected <class> category in this image.
26. Your help is required to generate distinct masks for each category of <class> in this image.
27. It would be appreciated if you could assist in creating separate masks for each <class> category in this image.
28. Let's collaborate on segmenting all categories falling under the <class> category in this image and generating masks.
29. Assisting me in generating distinct masks for each class categorized as <class> would be greatly appreciated.
30. Providing assistance in generating masks that accurately identify the categories falling under <class> in this image would be greatly helpful.

Table A20: **A list of instructions for semantic segmentation.**

1. Can you examine the image and pinpoint the keypoint locations of the <class>?
2. Could you analyze the picture and determine the keypoint placement of the <class>?
3. Please inspect the image and locate the keypoints for <class>.
4. Can you evaluate the photo and identify where the keypoints of <class> are situated?
5. Look at the image and detect the keypoint positions of the <class>.
6. Please analyze this image and find the keypoints of <class>.
7. Can you check the image and show me where the keypoints of <class> are located?
8. Please find the exact keypoint position of the <class>.
9. Please observe the photo and identify the keypoint locations of the <class>.
10. Can you review the image and point out the keypoints of <class>?

Table A21: **A list of instructions for pose estimation.**

1. Give me a concise description of the image. Answer the question and localize each object.
2. Please briefly summarize the content of this image. Answer the question and localize each object.
3. What does this picture show? Please summarize briefly. Answer the question and localize each object.
4. Can you give me a quick overview of what's depicted in this image? Answer the question and localize each object.
5. Could you describe the key elements in this photograph? Answer the question and localize each object.
6. Offer a brief explanation of what this image represents. Answer the question and localize each object.
7. Sum up the contents of this picture in one or two sentences. Answer the question and localize each object.
8. What is the main content in this image? Answer the question and localize each object.
9. Provide a brief caption for the picture. Answer the question and localize each object.
10. Could you give me a short description of the image? Answer the question and localize each object.

Table A22: **A list of instructions for grounded caption.**

1. Can you examine the image and segment the corresponding objects denoted as <regions>?
2. Where are the objects marked by <regions> in the image? Could you help me segment these objects?
3. Could you please segment all the corresponding objects according to the visual prompt as <regions>?
4. Can you help me draw the instance segmentation masks of <regions> in the picture?
5. Please help me find all the objects shown as <regions> and segment them.
6. I'd like to know the objects outlined by <regions>. Please help me draw their masks.
7. Given the <regions>, I need your help to segment the corresponding object masks.
8. Examine the image and identify all the objects that belong to the provided <regions>.
9. I'm interested in the objects labeled as <regions>. Could you please draw their instance masks?
10. There are some regions represented by <regions>. I need your assistance to find their corresponding objects.

Table A23: **A list of instructions for interactive segmentation.**

1. Generate image with caption: <caption>.
2. Can you give me the image with caption: <caption>.
3. Help me to generate this image: <caption>.
4. Generate the image according to the caption: <caption>.
5. According to the caption, generate the image: <caption>.
6. An image with caption: <caption>.
7. Can you visualize this caption: <caption>.
8. Create an image based on this caption: <caption>.
9. Generate a visual representation for this caption: <caption>.
10. Provide me with an image corresponding to this caption: <caption>.
11. Craft an image with the following caption: <caption>.
12. Generate an image accompanied by this caption: <caption>.
13. Turn this caption into an image: <caption>.
14. Generate an image reflecting this caption: <caption>.
15. Translate this caption into a visual representation: <caption>.
16. Produce an image that matches this caption: <caption>.
17. Create an image in line with this caption: <caption>.
18. Generate an image to illustrate this caption: <caption>.
19. Construct an image based on the given caption: <caption>.
20. Give me an image associated with this caption: <caption>.

Table A24: **A list of instructions for image generation.**

