# OpenReview forum: "VisionLLM v2: An End-to-End Generalist Multimodal Large Language Model for Hundreds of Vision-Language Tasks"
_NeurIPS.cc/2024/Conference — NeurIPS 2024 poster_

### Official Review · Reviewer_nisa · 2024-06-16

**Soundness:** 3
**Presentation:** 3
**Contribution:** 2
**Rating:** 5
**Confidence:** 4

**Summary:**

The paper proposes a model that supports a wide range of multimodal tasks, beyond text generation. The approach leverages an LLM, modality specific encoders and task specific decoders to effectively handle different tasks. The LLM communicates with task decoders via  “super link”,  which consists of special tokens and soft prompts for each task.

**Strengths:**

- The work is extensively evaluated on many benchmarks
- The model competes with other more specialized approaches
- The paper is well written and easy to follow
- The paper addresses an important problem, building efficiently a generalist models is still an open-research question

**Weaknesses:**

1. The paper title is a bit misleading. It gives the impression that the paper proposes a single model that can handle many tasks, while it is basically an agglomeration of many powerful pretrained models (CLIP, LLM, Stable Diffusion, UniPose…).
2. The contribution  is limited. The super link is basically soft prompts (learnable tokens) and special tokens per task. Soft prompt is widely used in other approaches (shared learnable query as in InstructBLIP) or (learnable query per task as in eP-ALM/MAPL). For the special tokens, FROMAGe uses special tokens to handle multimodal outputs. The main contribution of the paper is in scaling this approach to many multimodal tasks. It is also important for the paper to discuss these similar approaches.
3. The paper lacks details about the model training. In particular which training objectives are used on top of the task-specific decoders? And how the different losses are weighted?
4. Did the authors experiment with one-stage training instead of the 3-stages training? I didn’t find any experiment to support this design choice
5. The paper claims “Multimodal In-Context Ability. our model …  exhibits superiority over the previous in-context models …” However there is no quantitative comparison to support the superiority of the proposed model in a a few-shot ICL setting.

InstructBLIP: Dai, Wenliang, et al. "Instructblip: Towards general-purpose vision-language models with instruction tuning." Advances in Neural Information Processing Systems 36 (2024).

eP-ALM: Shukor, Mustafa, Corentin Dancette, and Matthieu Cord. "ep-alm: Efficient perceptual augmentation of language models." Proceedings of the IEEE/CVF International Conference on Computer Vision. 2023.

MAPL: Mañas, Oscar, et al. "Mapl: Parameter-efficient adaptation of unimodal pre-trained models for vision-language few-shot prompting." arXiv preprint arXiv:2210.07179 (2022).

FROMAGe: Koh, Jing Yu, Ruslan Salakhutdinov, and Daniel Fried. "Grounding language models to images for multimodal inputs and outputs." International Conference on Machine Learning. PMLR, 2023.

**Questions:**

Check weaknesses section (e.g. 3-4)

**Limitations:**

The authors discuss the limitations and the broader impact in the paper.

---

> ### Author Rebuttal · Authors · 2024-08-07
>
> Dear reviewer nisa,
>
> Thanks very much for your valuable comments. We hope our responses can address your concerns and clarify our contribution.
>
> **Q1: The paper title is a bit misleading because the model is an agglomeration of many powerful pretrained models instead of a single model.**
>
> **A1:** We suppose that a network that could be trained end-to-end is considered as a single model. In the deep learning community, It is a common practice to link the different pretrained models in one network, such as Flamingo, LISA, and GLaMM. We extend the one-to-one linking to the one-to-many linking, which still should be regarded as a single model, NOT an agglomeration of many powerful pretrained models.
>
>
> **Q2: Limited contribution regarding the soft prompt and special tokens.**
>
> **A2:** (1) InstructBLIP, ep-ALM, and MAPL use learnable queries (i.e., soft prompts) to connect the modality encoders and LLM. FROMAGe uses a special token for image-text retrieval so as to handle multimodal outputs, where the images are not generated from the network end-to-end. These works still remain constrained to **text-based outputs**. However, our work is significantly different from theirs in that we support hundreds of tasks by largely extending the output formats for MLLM, e.g., **box, mask, keypoint, text, and image**.
>
> (2) When scaling to various tasks across a broad range of domains, it requires us to address several challenges: (i) **precise decoder invocation**, (ii) **mitigating task conflicts**, and (iii) **efficient message transmission** in an end-to-end manner. Moreover, as the number of tasks increases, the difficulty of addressing these challenges will also significantly rise. Our proposed super link is a simple but non-trivial solution. It provides a new approache for connecting multiple decoders and coordinating hundreds of tasks, which have not been explored by previous works. Specifically, **first**, the super link contains a routing token that focuses on scheduling decoders. It can precisely determine which decode should be used after being trained with large-scale of corpus. **Second**, we use un-shared super-link queries for different decoders, which can mitigate the issue of task conflicts. **Last**, super-link queries can be easily accessed by their corresponding decoder and act as conditions for different decoders in solving various tasks.
>
> (3) Despite the simplicity of our super link, it is able to extend MLLMs to handle hundreds of tasks across various domains. Without redundant design, a simple yet effective method might provide more clear insights into this research topic. Therefore, we believe that our contributions to the study of generalists are clear and should not be overlooked. We will include the above discussion in the revised version.
>
> **Q3: Details about model training.**
>
> **A3:** (1) We have provided the model training details, e.g., training strategy and hyper-parameters, in Appendix D.
> Regarding the training objectives for task-specific decoders, we have given brief descriptions in Appendix B.2. Specifically, we keep the original losses for the decoders. Grounding-DINO optimizes the combination of classification loss, box losses (including l1 loss and gIoU loss),  and mask losses (including binary mask loss and DICE loss). The training objective for UniPose is the combination of classification loss, box losses (including l1 loss and gIoU loss), and keypoint losses (including l1 loss and OKS loss). For Stable-Diffusion and Instruct-Pix2Pix, we employ two MSE losses: one between the CLIP text features and mapping features, and the other one between the ground-truth noise and predicted noise.
>
> (2) We simply sum up all the losses from LLM and task-specific decoders without reweighting for each exponent, i.e., $L_{\text{total}} = L_{\text{llm}} + L_{\text{gdino}} + L_{\text{unipose}} + L_{\text{sd}} + L_{\text{ip2p}}$.
>
> Thanks to the reviewer for the suggestion.  We will add more detailed descriptions of training losses in the revised version.
>
>
> **Q4: Experiment with one-stage training instead of three-stage training.**
>
> **A4:** Please refer to common questions Q2 for the experiments and explanations. We will add this part in the revised version.
>
> **Q5: Quantitative comparison for the in-context learning setting.**
>
> **A5:** (1) In-context segmentation
>
> To fairly compare with state-of-the-art methods, e.g., Painter [a1] and SegGPT [a2], we construct a benchmark based on the validation set of COCO2017, where the number of in-context examples used during inference ranges from 1 to 5. The results are shown in the following table.
>
> | Method | mIoU |
> | :----: | :----: |
> | Painter [a1] | 44.26 |
> | SegGPT [a2] | 54.25 |
> | VisionLLM v2| 68.15 |
>
> (2) In-context image captioning
>
> We follow the same evaluation protocol as OpenFlamingo [a3] and use 4-shot to assess the performance between different methods. The validation set is built upon COCO2017. We present the results in the following table.
>
> | Method | METEOR | CIDEr |
> | :----: | :----: | :----: |
> | OpenFlamingo [a3] | 13.80 | 104.61 |
> | VisionLLM v2| 18.56 | 152.74 |
>
> Generally, VisionLLM v2 exhibits clear performance advantages compared with state-of-the-art methods in both in-context learning settings, which demonstrates the superior in-context capacities of our method. These results will be updated in our final paper.
>
> References
>
> [a1] Wang, Xinlong, et al. "Images speak in images: A generalist painter for in-context visual learning." Proceedings of the IEEE/CVF Conference on Computer Vision and Pattern Recognition. 2023.
>
> [a2] Wang, Xinlong, et al. "Seggpt: Segmenting everything in context." arXiv preprint arXiv:2304.03284 (2023).
>
> [a3] Awadalla, Anas, et al. "Openflamingo: An open-source framework for training large autoregressive vision-language models." arXiv preprint arXiv:2308.01390 (2023).

---

> > ### Comment · Reviewer_nisa · 2024-08-10
> >
> > Thanks for detailed feedback. The reviewers addressed most of my concerns. I am still not convinced about the originality of the work. However, the work contains some important messages that could benefit the community. Thus, I will raise my score.

---

> ### Author Response · Authors · 2024-08-10
> **Official Comment by Authors**
>
> Thank you so much for taking the time to re-evaluate our paper, thoughtfully considering our responses, and agreeing to raise the score. We will carefully take your valuable suggestions and continuously improve our paper.

---

### Official Review · Reviewer_W77r · 2024-07-06

**Soundness:** 3
**Presentation:** 3
**Contribution:** 3
**Rating:** 7
**Confidence:** 5

**Summary:**

This paper introduces an advanced multimodal large model (MLLM) that integrates visual perception, understanding, and generation in a unified framework. Unlike traditional models limited to text outputs, it expands its capabilities to tasks like object localization, pose estimation, and image generation and editing via a "super link" mechanism to connect the MLLM with task-specific decoders, facilitating flexible information and gradient feedback transmission while resolving multi-tasking training conflicts. The model is trained on data from hundreds of vision and vision-language tasks, allowing it to generalize across these tasks with shared parameters and achieve performance comparable to task-specific models. VisionLLM v2 aims to enhance the generalization of MLLMs.

**Strengths:**

1. The starting point of the paper is very good: enabling LLMs to use a variety of tools so that the model can handle various tasks.
2. Model's performance is excellent, reaching the state-of-the-art for general models across various tasks.
3. The experiments are extensive, involving a significant amount of engineering work, and integrating various tasks and datasets.

**Weaknesses:**

1. The training process is relatively complex, consisting of three stages, making it difficult to promote and use in the industry. The method described in the paper seems like it could also be done in a single end-to-end training stage, similar to LLMs. [1] is the one-stage training general vision model. Discussing the general models [1][2]  trained in a one-stage manner with a pure language interface would be better. :)
2. In Table 3, the detection performance is basically on par with the single-task decoder performance later on. Does this imply that the performance mainly depends on the task-specific decoder? Is the method proposed in the paper primarily aimed at facilitating the integration and scheduling of various decoders? Will the computation of MLLM enhance the performance of the subsequent decoder?
3. The advantage of LLMs is not only their compatibility with various tasks but also their ability to mutually enhance different tasks, which is the so-called multi-task capability. In the method described in the paper, different tasks share the same MLLM and encoder. So, can joint training of different tasks, like GiT[1], lead to mutual enhancement?

Points 2-3 are just areas where I think there might be room for improvement, but they won't significantly affect my decision on whether to accept this paper. It's just for discussion. :)

[1] GiT: Towards Generalist Vision Transformer through Universal Language Interface. (ECCV2024)

[2] Fuyu-8B (blog)

**Questions:**

This question is just to hear the authors' perspectives: Should a general model use a bridging approach (LLaVa or VisionLLM) or a simple multi-layer transformer (Fuyu8B, GiT, Gemini)? The former might be easier to implement but has complex training and scaling issues, while the latter has greater advantages in scaling and simpler training, similar to GPT. (Authors can answer freely without considering that the paper uses a bridging framework. This will not affect the final score.) :)

**Limitations:**

This is purely academic research and does not involve any potential negative societal impact.

---

> ### Author Rebuttal · Authors · 2024-08-07
>
> Dear reviewer  W77r,
>
> Thanks a lot for your insightful reviews and support for our work! We hope our responses can address your questions.
>
>
> **Q1:  Complex training process.**
>
> **A1:** We kindly invite the reviewer to refer to the common question Q2 for the details.
> GIT [1] mainly focuses on visual tasks and uses a smaller scale of datasets (i.e., COCO, RefCOCO series, COCO caption, ADE20K) for training. And GIT is basically evaluated on these datasets without evaluating its conversation ability. Fuyu-8B [2] is a decode-only transformer that is trained for image-level visual question answering. These two works focus on a single task or related visual tasks, making their training can be completed in one stage.
>
> **Q2: Does the performance mainly depend on the task-specific decoder? Does the proposed method aim at facilitating the integration and scheduling of various decoders? Will MLLM enhance the performance of decoders?**
>
> **A2:** (1) The performance largely depends on the task-specific decoders. We propose the end-to-end framework to increase the openness of decoders. For example, we can extend the capacity of Grounding-DINO by adapting to more profound categories, more diverse domains, and more flexible textual conditions.
>
> (2) The answer to the second question, "Does the proposed method aim at facilitating the integration and scheduling of various decoders?", is yes. We integrate several task-specific decoders in one network, and use MLLM to select the task-specific decoder properly.
>
> (3) The advantage coming from MLLM is its strong language processing ability for interpreting and reasoning over the user's prompt.  This endows the model with greater textual perception and sentence reasoning capabilities, thereby enhancing the performance of tasks that rely on openness. We have conducted the ablation experiments by replacing the LLM from Vicuna-7B with stronger InternLM-20B. We find that for the grounding task on RefCOCO, the model obtains the significant +2.8 points on P\@0.5, which meets our expectations. While for object detection on COCO, we do not see performance improvement. This is because the traditional language model, e.g., BERT, is sufficient for encoding the category information.
>
> **Q3: Can joint training of different tasks lead to mutual enhancement?**
>
> **A3:** Please refer to common question Q1 for the details. We use different decoders to complete various tasks,  and we do not observe obvious mutual enhancement among tasks in this work.
>
> **Q4: Should a general model use a bridging approach (LLaVA or VisionLLM) or a simple multi-layer transformer?**
>
> **A4:** Very good question. From our perspective, the bridging approaches are easier to implement, in order to enhance the LLM with various capabilities beyond the text output.
>
> We agree with the reviewer that the simple multi-layer transformer architecture is more unified and concise and has advantages in simpler training. However, at the current technology level, there are many tasks and modalities in the real world that are hard to model using the next token prediction. For example, there are still some open research questions: Do image patches follow the left-to-right order? How to effectively represent structural outputs such as segmentation masks and protein structure?
>
> In general, we believe it is necessary to integrate specialized models into general models to advance the field of generalist models in the short term.

---

> > ### Comment · Reviewer_W77r · 2024-08-13
> >
> > Thank you for the author's response. Regarding Q1, I'm sorry to say that I am not yet fully convinced. A single-stage training framework is essential for industrial applications; otherwise, the final paper might remain confined to academia. The frameworks that can be widely adopted in the industry are those that are simple and effective. Even if a single-stage training approach may reduce performance, the industry is likely to prefer it. We all hope to create work that is truly useful for the industry, not just a paper. Therefore, I hope the author will discuss this limitation in the camera-ready version, comparing it with single-stage works like GiT and Fuyu-8B. Looking forward to seeing the author's future work on refining and optimizing the training strategy.

---

> > > ### Author Response · Authors · 2024-08-14
> > > **Official Comment by Authors**
> > >
> > > Thank you for your thoughtful feedback. We design the three-stage training strategy in this work to effectively train the model from an initial vision encoder and a large language model, with the primary goal of maximizing performance.
> > >
> > > We agree that it is meaningful for training simplicity and efficiency. As mentioned in common questions Q2, our model has the potential to be trained in a single stage by adjusting the dataset sample ratios. And we believe the performance gap between the one-stage and three-stage strategies could be significantly reduced by starting with pretrained, strong vision-language models (e.g. LLaVA, InternVL). In our recent exploration, we directly load the pretrained InternVL and augment it with Grounding-DINO. We meticulously adjust the dataset ratios of all combined datasets to simulate 1 epoch training on chat data and 12 epoches training on detection data. Then the model is trained within one stage to simplify the training pipeline. Through this careful configuration, we observe the much smaller performance decrease: -0.5 / -2.0 points on MMB EN/CN compared with original InternVL, -1.3 points for object detection on COCO compared with original Grounding-DINO.
> > >
> > > In the final version of our paper, we will include a detailed discussion on the limitations of our approach compared to single-stage works like GIT and Fuyu-8B, highlighting the trade-offs between performance and training simplicity.
> > >
> > > Thank you once again for your valuable feedback. We will incorporate these considerations into our future research to explore simpler training strategy, with the hope that our work will ultimately bridge the gap between academic research and industrial application.

---

### Official Review · Reviewer_x7n8 · 2024-07-12

**Soundness:** 3
**Presentation:** 3
**Contribution:** 3
**Rating:** 5
**Confidence:** 4

**Summary:**

This paper proposes an end-to-end generalist Multimodal Large Language Model (MLLM) for a variety of vision-language tasks, including captioning, detection, segmentation, and image generation. It introduces the concept of "Super Link" for triggering different tasks and attaches corresponding task decoders to the MLLM through query embeddings for end-to-end gradient updates. The model demonstrates strong results on the problems of interest and opens a new avenue for future multimodal generalist studies.

**Strengths:**

The paper is well-written, and the idea is well-motivated. The evaluation results are generally sufficient.

**Weaknesses:**

The paper lacks a comprehensive review of related work. Relevant works such as AnyGPT [r1], Chameleon [r2], the CM3 series, and other earlier publications that unify image generation in MLLM with image quantization and decoding should be reviewed and acknowledged.

Clarification is needed on how super-link queries are generated. For [DET] and [SEG], these can be boilerplate templates, but what about text-to-image generation? Are the super-link queries generated on-the-fly based on the context? Examples for [GEN], similar to the one at the bottom of Page 5, would be appreciated.

Regarding the joint multi-task training stage elaborated in Sec. 3.3, ablation studies on how each task benefits the others should be conducted. Does performing fine-grained vision tasks enhance the model's image generation capability, or do multimodal understanding and generation remain conflicting as shown in the literature? Do detection/segmentation tasks lead to better controllability or spatial reasoning for image generation? These questions need to be studied.

[r1] Zhan et al., AnyGPT: Unified Multimodal LLM with Discrete Sequence Modeling.

[r2] Chameleon Team, Chameleon: Mixed-Modal Early-Fusion Foundation Models.

**Questions:**

see previous sections

**Limitations:**

appear to be sufficient

---

> ### Author Rebuttal · Authors · 2024-08-07
>
> Dear reviewer x7n8,
>
> Thanks so much for your constructive comments and support for acceptance. We hope our responses can address your concerns.
>
> **Q1: More related works.**
>
> **A1:** Thanks for pointing out the missing related works. AnyGPT builds a multimodal text-centric dataset for any-to-any multimodal generation (text, image, speech, music) with sequence modeling. Chameleon uses fully token-based representations for both texts and images, capable of understanding and generating interleaved image-text sequences. CM3 series are autoregressive models for text-to-image and image-to-text tasks. All these works could unify image understanding and generation in one network. Our model can support more vision and vision-language tasks.
> We will respectfully cite these works and add the discussion to the related work section of our paper.
>
>
> **Q2: Clarification on how super-link queries are generated. Template example for text-to-image generation.**
>
> **A2:** (1) Here is a template example for the text-to-image generation:
>
> ```
> USER: Please generate an image with caption: a dog near the sea.
> ASSISTANT: Sure, here it is [GEN].
> ```
>
> We will add this example in our revised version.
>
> (2) In the following, we would like to give clearer clarification about the generation of super-link queries.
>
> The super-link queries are initialized as learnable weights `nn.Embedding`. During inference, whenever the LLM predicts the special token such as [DET], [GEN], the super-link queries will be automatically appended after it.  This means that, in the current generation step, the size of input embeddings for LLM is expanded from ``[1, c]`` to ``[1 + num_embeds, c]``, where `num_embeds` is the number of super-link queries, `c` is the hidden size of LLM. So, it is true that the super-link queries are generated on-the-fly based on the context.
>
> **Q3: Study on how each task influences the others.**
>
> **A3:** We have provided detailed responses in the common questions Q1. Based on the current experimental results, we have not yet observed the positive impact of vision tasks on image generation.

---

> ### Comment · Area_Chair_dnwd · 2024-08-12
> **Concerns addressed?**
>
> Dear reviewer, the authors have responded to your questions - are you satisfied by their response? If so, would you like to update your rating? Is there any more information you need from the authors to make that decision?

---

> > ### Comment · Reviewer_x7n8 · 2024-08-12
> > **Questions have been addressed in the rebuttal**
> >
> > The rebuttal has addressed the questions from the initial review and the authors indicated corresponding revisions in the final version. Therefore, I maintain my original positive review.

---

> > > ### Author Response · Authors · 2024-08-13
> > > **Thanks for your positive feedback**
> > >
> > > We are glad we could address your questions. We sincerely appreciate the time and effort you put into reviewing our paper and providing valuable comments. We will ensure the revisions are incorporated into the final version.

---

### Author Rebuttal · Authors · 2024-08-07

Dear all reviewers and ACs,

We sincerely thank you for all the time and effort in reviewing our paper and giving valuable comments. We are really encouraged that all the reviewers appreciate the good motivation, extensive experiments, strong performance, and clear representations. We will first answer the common questions, and then give responses to each reviewer separately. Hope our responses can address your concerns.
We are happy to further discuss with you if there are still other concerns. Thanks for helping improve our paper.

&nbsp;

### Common questions

**Q1: Multi-task benefiting.**

**A1:** (1) As indicated by previous works [a1, a2], different tasks with shared parameters may cause conflict with each other. This is mainly due to inconsistent optimization in multi-task learning. To validate this, we start from the same checkpoint and train the model on a single task (image VQA, instance segmentation, or image generation) for 1000 iterations. Then, we record the loss change for all three tasks. The results are presented in the following table, where the first column represents the training tasks, and the first row represents the testing tasks.

| Train \ Test | Image VQA | Inst Seg.  | Image Gen. |
| :----: | :----: | :----: | :----: |
| **Image VQA** | - 0.01 |  - 0.11 | - 0.04 |
| **Inst Seg.** | + 0.04 | - 0.12 | + 0.19 |
| **Image Gen.** | + 0.03 | + 0.02 | - 0.04 |

In the table, a decrease in the loss value indicates beneficial training for the task, while an increase is detrimental. We can observe that training on image visual question answering (VQA) is advantageous for all three tasks, which is reasonable as the conversation ability of MLLM is enhanced. Whereas training exclusively on instance segmentation or image generation leads to conflicts with other tasks. Uniperceiver-MoE [a1] has concluded that task conflicts are more significant when closer to the output layer, which aligns with our findings.

(2) In our model, using different decoders did not reveal complementary effects during the multi-task learning. Therefore, we employ un-shared super-link queries to address task conflicts. Using shared super-link queries, on the other hand, can even lead to a decrease in performance, as illustrated in Figure 4 of the paper.

(3) Although integrating different decoders in one network may lead to conflicts, we would like to claim the importance of building one end-to-end generalist MLLM: (i) enhance the capacities of MLLM to complete different tasks beyond the text outputs. (ii) increase the openness of decoders to adapt to more diverse domains and more flexible text instructions.

References

[a1] Zhu, Jinguo, et al. "Uni-perceiver-moe: Learning sparse generalist models with conditional moes." Advances in Neural Information Processing Systems 35 (2022): 2664-2678.

[a2] Yu, Tianhe, et al. "Gradient surgery for multi-task learning." Advances in Neural Information Processing Systems 33 (2020): 5824-5836.


**Q2: One-stage v.s. three-stage training.**

**A2:** (1) The intrinsic reason for the design of three-stage training is to ensure the conversation ability of the multimodal large language model (MLLM) while extending its other capabilities. However, this introduces a training conflict: the MLLM requires only 1 epoch of training on chat data to prevent overfitting, whereas the decoders need longer training epochs (e.g., Grounding-DINO need 12 epochs of training on visual data) to achieve convergence. Thus, we designed the three-stage training strategy: Stage-1 to obtain an MLLM with strong conversation ability, Stage-2 to finetune the MLLM to acquire basic additional capabilities, and Stage-3 to train the decoders only until convergence.

(2) One possible solution for one-stage training is to give a higher sample ratio for the visual data. In the following, we conduct the ablation study to study the effect of one-stage v.s. three-stage training. We use image-level chat data, COCO, and COCO-Pose for image understanding, instance segmentation, and pose estimation, respectively. For one-stage training, we repeat the COCO and COCO-Pose datasets 12 times. The three-stage training undergoes the same process as specified in the paper.

| | TextVQA | MME | MMB EN/CN | COCO | COCO-Pose |
| :----: | :----: | :----: | :----: | :----: | :----: |
| one-stage | 53.2 | 1284.4 | 61.9 / 51.4 | 54.9 / 44.6 | 74.1 |
| three-stage | 66.2 | 1507.1 | 77.8 / 68.5 | 56.3 / 47.6 | 74.2 |

As can be seen from the table, the conversation ability of the model is significantly decreased due to extreme data imbalance. And the performance of instance segmentation and pose estimation is also slightly reduced. These results prove the effectiveness of the three-stage training.

(3) We would like to emphasize that training a generalist model to support various tasks (multimodal conversation, object segmentation, pose estimation, image generation end editing, etc.) without performance degeneration is a significant challenge in our work, especially considering that we leverage large-scale datasets from various domains. It is extremely difficult to achieve optimal performance for all the tasks from a broad range of domains at the same training point. We dedicated substantial effort to address this issue, ultimately developing the three-stage training strategy.

---

### Decision · Program_Chairs · 2024-09-25

**Decision:**

Accept (poster)

**Comment:**

The work presents a Vision-Language model for producing diverse output types like text, bounding boxes, segmentations, pose, segmentation keypoints etc. The key novelty is a "super-link" mechanism based on learnable soft-prompts for connecting a multimodal LLM to various task-specific decoders.

The work received positive scores (A, BA, BA) from reviewers. The reviewers appreciated extensive experiments, good performance, and presentation/writing. Concerns raised by reviewers included complexity of the 3 stage training pipeline and transfer between tasks. The authors rebuttal appears to have satisfied these reviewer concerns.

Having read the reviews, rebuttal, and the discussion, the AC generally agrees with the reviewers and is happy to recommend "accept". The AC recommends the authors to incorporate the discussion in their revised draft and also to refer to the figures and tables in the appendix in the main paper (for example, would be useful to refer the reader to Fig A13 in the caption of Fig 2 in the main paper). The AC also found Fig.1 to be too abstract and not be very informative and would suggest the authors modify it to ground it more to the proposed approach (for examples, its unclear from the figure what the difference between a,b,c are and how exactly each of them is implemented).